# Freeze, Prompt, and Adapt: A Framework for Source-free Unsupervised GNN Prompting

**Peyman Baghershahi**                                                    *pbaghe2@uic.edu*
*Department of Computer Science*
*University of Illinois Chicago*

**Sourav Medya**                                                         *medya@uic.edu*
*Department of Computer Science*
*University of Illinois Chicago*

**Reviewed on OpenReview:** *https://openreview.net/forum?id=9KKgIQwCLO*

## Abstract

Prompt tuning has become a key mechanism for adapting pre-trained Graph Neural Networks (GNNs) to new downstream tasks. However, existing approaches are predominantly supervised, relying on labeled data to optimize the prompting parameters and typically fine-tuning a task-specific prediction head—practices that undermine the promise of parameter-efficient adaptation. We propose *Unsupervised Graph Prompting Problem (UGPP)*, a challenging new setting where the pre-trained GNN is kept entirely frozen, labels on the target domain are unavailable, the source data is inaccessible, and the target distribution exhibits covariate shift. To address this, we propose UGPROMPT, the first fully unsupervised GNN prompting framework. UGPROMPT leverages consistency regularization and pseudo-labeling to train a prompting function, complemented with diversity and domain regularization to mitigate class imbalance and distribution mismatch. Our extensive experiments demonstrate that UGPROMPT consistently outperforms state-of-the-art supervised prompting methods with access to labeled data, demonstrating the viability of unsupervised prompting as a practical adaptation paradigm for GNNs. Code: https://github.com/pbaghershahi/UGPrompt.

## 1 Introduction

Prompt tuning (Li & Liang, 2021; Lester et al., 2021) has driven many recent developments for Large Language Models (LLMs). These methods optimize external prompting parameters to guide a model's responses. The goal is to avoid fine-tuning the model's vast number of internal parameters for adaptation to new downstream tasks, keeping the core pre-trained knowledge intact (Zheng et al., 2025; Han et al., 2024; Liu et al., 2023a). However, prompting becomes more challenging for graphs because: First, there is a wide range of graph tasks; thus, unlike general language tasks (Devlin et al., 2019; Radford & Narasimhan, 2018), *task unification* is difficult in graphs. This hinders the collection of large-scale data, e.g., from the internet (Radford et al., 2019), creating bottlenecks in pre-training *large* Graph Neural Networks (GNNs) for general-purpose use. Second, because of the limited reasoning capabilities of GNNs (Morris et al., 2024; Mao et al., 2024a), it is non-trivial to constitute instructive graph prompts in human (natural) language (Fatemi et al., 2024). Thus, GNN prompting should be understood as a parameter-efficient adaptation paradigm inspired by LLM prompting, rather than as evidence that current GNNs possess LLM-like zero-/few-shot reasoning abilities.

A few recent studies have adopted prompting for GNNs to align the objectives of pre-training on source data and fine-tuning on target data, mostly following the "pre-train, prompt, fine-tune" pipeline (Sun et al.,

2022). These works design unified tasks that allow optimizing a GNN with semantically similar objectives on the pretext and downstream tasks (Huang et al., 2023; Liu et al., 2023b; Fang et al., 2023; Sun et al., 2023; Yu et al., 2024b; Chen et al., 2025). However, the current GNN prompting paradigm suffers from two key limitations that hinder its efficiency relative to LLMs. First, existing methods rely heavily on labeled data — costly to obtain — to achieve competitive performance. Second, they require training new projection heads for each downstream task, a form of *lightweight fine-tuning* (Li & Liang, 2021). These dependencies on additional parameters and labeled data (especially in scenarios where the original source data is inaccessible, e.g., due to privacy) prevent GNNs from being used as truly frozen models. This gap motivates our work to establish a more practical and efficient prompting paradigm for GNNs.

To directly address these limitations, we first introduce the Unsupervised Graph Prompting Problem (UGPP), a rigorous novel problem formulation. The UGPP setup evaluates a method under four key conditions: the GNN's parameters are frozen, there is a covariate shift in the target data distribution, no target labels are available for adaptation, and the source data is inaccessible. While this setup shares similarities with Unsupervised Source-Free Domain Adaptation (SFDA) (Li et al., 2024), it differs in a crucial aspect: UGPP requires the entire source-trained GNN to be frozen, whereas SFDA methods (Mao et al., 2024b; Zhang et al., 2024) rely on fine-tuning the model's parameters. This setting places our work within the paradigm of parameter-efficient prompting, rather than full model adaptation.

Within this challenging setup, we propose UGPROMPT, the first fully unsupervised GNN prompting framework. UGPROMPT trains a prompting function using consistency regularization and confident pseudo-labeling, enabling the frozen GNN to adapt its knowledge to the new target distribution. To ensure robustness, we introduce two additional regularization techniques: one to counteract prediction bias from class imbalance and another to make the prompted graphs close to the original data distribution. Our extensive experiments show that UGPROMPT, despite being fully unsupervised, consistently outperforms state-of-the-art (SOTA) prompting methods that have the advantage of full access to labeled data. Our major contributions are summarized as follows.

- **Problem formulation.** We propose UGPP, a novel problem setup that isolates the true effectiveness of a prompting function by disallowing any updates to the source-trained GNN's parameters.

- **Novel unsupervised methodology.** We propose UGPROMPT, the first fully unsupervised GNN prompting method that leverages consistency regularization and pseudo-labeling to adapt a frozen GNN to new data distributions.

- **Empirical analysis.** We demonstrate that UGPROMPT substantially outperforms supervised SOTA methods on node and graph classification tasks, validating the effectiveness of unsupervised adaptation in this novel and more practical setting.

## 2 Background & Problem Formulation

Previous studies on adapting prompt tuning for GNNs use lightweight fine-tuning (Li & Liang, 2021; Lester et al., 2021) with supervision, which has been addressed widely in different domains such as computer vision (van den Oord et al., 2018; Chen et al., 2020; Zhuang et al., 2021) and NLP (Devlin et al., 2019; Ruder et al., 2019). Specifically, recent works have followed the *"pre-train, prompt, fine-tune"* setup (Sun et al., 2022; Liu et al., 2023b; Sun et al., 2023; Fang et al., 2023). In this section, we first introduce the setting and discuss its limitations, and then present our proposed problem setting, which addresses these limitations.

### 2.1 Pre-train, Prompt, Fine-tuning

This pipeline aims to bridge the generalization gap between pre-trained GNNs and downstream tasks that differ semantically from the pretext tasks. Unlike the traditional supervised and "pre-training, fine-tuning" methods, this pipeline employs a task unification step before pre-training and fine-tuning. This is essential because it aligns the pretext and downstream objectives to optimize a pre-trained model on new datasets. The steps are as follows.

*(1) Pre-train.* Formally, given a set of tuples $\mathcal{S} = \{(\mathcal{T}_i, \mathcal{D}_i)\}_{i=1}^{N_s}$, with $N_s$ samples, from pretext task $\mathcal{T}_i$ and dataset $\mathcal{D}_i$, first all the tasks are unified to task $\mathcal{T}_u$ and the corresponding changes apply for their associated datasets to make a new set $\mathcal{S}_u = \{(\mathcal{T}_u^i, \mathcal{D}_u^i)\}_{i=1}^{N_s}$. Then a GNN encoder $g(.;\theta_g)$ is pre-trained on $\mathcal{S}_u$ using an unsupervised approach such as contrastive learning (You et al., 2020; Xia et al., 2022). For downstream tasks, by adding a projection head $h(.;\theta_h)$ after the encoder, a model $\psi = h \circ g$ is formed, of which the encoder parameters $\theta_g$ are frozen, and only the head parameters $\theta_h$ will be trained. *(2) Prompting.* At this stage, a prompting function $f(., \theta_f)$ is employed to construct a prediction model $\varphi$. The prompting function is either a prefix module, i.e., $\varphi = h \circ g \circ f$, or a postfix one, i.e., $\varphi = f \circ h \circ g$. *(3) Fine-tuning.* In the final step, the set of parameters $\{\theta_h, \theta_f\}$ of $\varphi$ are optimized for every unified downstream task $\mathcal{T}_u$ with the new labeled samples.

*Limitations.* A deeper look at the *"pre-train, prompt, fine-tune"* pipeline reveals that although the current methods impose less trainable parameters compared to full fine-tuning (fine-tuning both pre-trained feature encoder and decoder), they involve partial (lightweight) fine-tuning (Han et al., 2024; Li & Liang, 2021; Lester et al., 2021), as they also train the GNN's decoder (projection head) along with the new parameters of the prompting function. Subsequently, labels become essential for this partial fine-tuning, and they are unable to leverage unlabeled data from large datasets when collecting labeled data is challenging (Radford et al., 2019). Also, fine-tuning a large pre-trained model on large datasets may introduce noisy information when the labeled downstream datasets are small (Bousquet et al., 2004; Shalev-Shwartz & Ben-David, 2014). Therefore, it reduces model generalization across diverse applications (Brown et al., 2020).

## 2.2 Our Problem Setting

To address the above limitations, we first need a suitable problem setting that offers insights into how well a prompting method performs when there is lack of labeled data. It is also crucial to evaluate if the method generalizes across tasks without fine-tuning the base GNN model parameters.

**Unsupervised Graph Prompting Problem (UGPP).** *Suppose a GNN model $\varphi(.;\theta_g, \theta_h) = h(.;\theta_h) \circ g(.;\theta_g)$ is given, where $g$ and $h$ are its encoder and decoder. Also, $\varphi$ is trained for task $\mathcal{T}$ on a labeled source dataset $\mathcal{D}_s = \{(x_s^i, y_s^i) : x_s^i \sim \mathcal{P}_X^s, \; y_s^i \sim \mathcal{P}_{Y|X}^s\}_{i=1}^{Ns}$, where $x_s^i$ is a sample (e.g., a graph or node), and $y_s^i$ is its associated label. The problem is to train a prompting module $f(.;\theta_f)$ on an unlabeled target dataset $\mathcal{D}_t = \{x_t^j : x_t^j \sim \mathcal{P}_X^t\}_{j=1}^{N_t}$, s.t. $\mathcal{P}_X^s \neq \mathcal{P}_X^t$, to enhance the performance of $\varphi$ for task $\mathcal{T}$ on $\mathcal{D}_t$, assuming that $\theta_g$ and $\theta_h$ are fixed, $\mathcal{D}_s$ is unobservable, and $\mathcal{P}_{Y|X}^t$ remains invariant across domains, i.e., $\mathcal{P}_{Y|X}^t = \mathcal{P}_{Y|X}^s$.*

This problem, UGPP, focuses on prompting without labeled data. A prompting method that performs well in this setting has three advantages: *First,* it is model- and task-agnostic, i.e., it works for conventional GNNs (e.g., GCN, GAT) and tasks (e.g., node/graph classification). *Second,* it is unsupervised, allowing the use of many datasets to improve generalization. *Third,* this setting does not depend on the source data. This is particularly beneficial when the source data is inaccessible, e.g., due to privacy issues. Throughout the paper, we use "source-trained" to refer to the model trained on the labeled source domain for task $\mathcal{T}$, and reserve "pre-trained" for the general prompting literature.

It is noteworthy that UGPP can be viewed as a restrictive prompt-based instance of Unsupervised SFDA. Conventional SFDA methods typically fine-tune the source-trained model, whereas UGPP keeps the source-trained GNN, including both the encoder and the prediction head, entirely frozen. Therefore, all adaptation is isolated in the external prompting function. This restriction allows us to evaluate the effect of prompting without entangling it with the base model's latent fine-tuning, and it also makes the setting model-agnostic. Please find additional discussions on the UGPP definition in Appendix A.4 We also compare our proposed method against recent graph SFDA methods in Appendix A.6.4

## 3 Our Method: UGPrompt

Here, we introduce UGPROMPT, an unsupervised GNN prompting method, to address UGPP.
**Motivation.** We aim to design an unsupervised prompting framework that helps a source-trained GNN make robust predictions while its parameters are frozen. To achieve this, we take advantage of pseudo-labeling

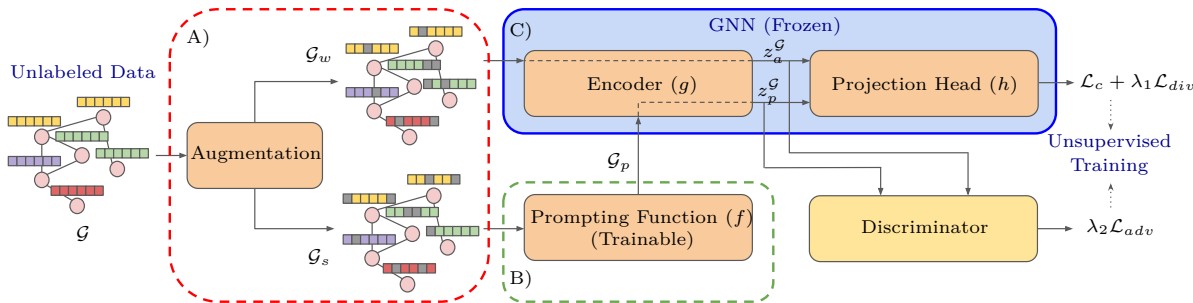

Figure 1: Overview of UGPROMPT. A) A non-parametric algorithm generates a weak augmentation $\mathcal{G}_w$ and a strong augmentation $\mathcal{G}_s$ from an unlabeled graph $\mathcal{G}$. B) The learnable prompting function $f$ generates a prompted graph $\mathcal{G}_p$ from $\mathcal{G}_s$. C) The base GNN with frozen parameters scores $\mathcal{G}_p$ and $\mathcal{G}_w$. A discriminator taking input from the latent representation ($z_a^{\mathcal{G}}$ for $\mathcal{G}_w$ and $z_p^{\mathcal{G}}$ for $\mathcal{G}_p$) of the GNN's encoder regularizes the model to adapt to the input distribution.

(Lee, 2013; Xie et al., 2020) and consistency regularization (Sajjadi et al., 2016; Laine & Aila, 2017). To train a prompting function in a fully unsupervised manner, UGPROMPT first obtains randomly augmented graphs from the target dataset. Then we employ consistency regularization (Zhang et al., 2021; Wang et al., 2023), where certain predictions of the base GNN are filtered by a confidence threshold and exploited as pseudo-labels for optimization. Indeed, knowledge from the source dataset is used to reduce prediction entropy and handle distribution shifts by leveraging GNN's confident predictions for unlabeled target data. The prompting module enhances target input samples by adding key information, making them more similar to the source samples. Empirical evidence to support this claim is in the Appendix A.7.1. We describe our framework (Figure 1) for graph-level tasks. Note that it also generalizes to node- and edge-level tasks.

**Overview of UGPrompt.** Our framework to address UGPP has a training and an inference step. The training step involves two components: *consistency-based prompting* and *prompt regularization*. *Consistency-based prompting* starts by augmenting each input graph twice, with one of the augmentations modified by a prompting function. Next, the GNN scores both samples. The objective is to train the prompting function such that the GNN produces "consistent prediction scores" for both samples of this pair with a certain confidence. In *prompt regularization* we introduce two regularization techniques: one to tackle biased predictions caused by a class imbalance in the data and another to prevent generating out-of-distribution (OOD) prompted graphs. During inference, a test graph is fed to the prompting function without augmentation, and its output goes to the GNN. We discuss a task unification step that generalizes to other graph tasks, e.g., node classification, in Appendix A.5.3.

## 3.1 The Consistency-based Prompting

Our prompting method is designed to reduce the discrepancy of the base GNN predictions over random augmentations of the same input graphs. We achieve this without labels by first augmenting the data with an algorithmic step, then generating pseudo-labels from the unlabeled augmentations when their assigned scores by the GNN meet a certain confidence threshold. We provide the details of these two stages below.

*Algorithmic augmentation.* Consistency regularization techniques (Zhang et al., 2021) train *strong augmentations* of samples using the pseudo-labels derived from their *weak augmentations*. Since our focus is on optimizing the prompting module rather than fine-tuning the base GNN, we adopt this technique as follows. We use a random non-parameterized augmentation algorithm (we use random feature masking in our experiments). More specifically, we mask a group of features with a certain probability. We augment an input graph $\mathcal{G}$ to create a weak augmentation $\mathcal{G}_w$ with masking probability $p_w$ and also a strong augmentation $\mathcal{G}_s$ with probability $p_s$, where $p_s > p_w$. We pass $\mathcal{G}_s$ through a learnable prompting function $f(.;\theta_f)$ to obtain a *prompted graph* $\mathcal{G}_p = f(\mathcal{G}_s; \theta_f)$. We keep $\mathcal{G}_w$ unchanged and call it a *non-prompted* augmentation graph.

*Learnable prompting.* We use a prefix prompting module to transform input samples for the base GNN. Our approach is generic enough to allow the integration of different prompting functions. For our

experiments, we choose a function $f$ that enriches node feature vectors, as this design aligns with our random feature masking augmentation technique. This way of adding learnable parameters is used in GPF-Plus (Fang et al., 2023). We discuss the choice of input-level prompt tuning and its relation to other parameter-efficient adaptation methods such as LoRA (Hu et al., 2022), in Appendix A.7.6.

Specifically, we learn a prompting function $f$ with parameter set $\theta_f = \{t_j^* : t_j^* \in \mathbb{R}^d\}_{j=1}^{n_t}$. For input graph $\mathcal{G}$ of $N$ nodes with nodes features set $X = \{x_i : x_i \in \mathbb{R}^d\}_{i=1}^{N}$, the function $f$ makes prompted graph $\mathcal{G}_p$ with feature set $X_p = \{x_i + t_i : x_i \in X, \ x_i, t_i \in \mathbb{R}^d\}_{i=1}^{N}$ such that $t_i = \sum_{j=1}^{n_t} \alpha_{i,j} t_j^*$ and $\alpha_{i,j} = \frac{\exp(x_i^T t_j^*)}{\sum_{l=1}^{n_t} \exp(x_i^T t_l^*)}$ .

***Consistency-based Objective.*** We optimize $\theta_f$ to minimize the discrepancy between the GNN's prediction scores for the non-prompted augmented graph $\mathcal{G}_w$ and the prompted graph $\mathcal{G}_p$. This would lower the entropy of the GNN's scores for the target unlabeled data. Intuitively, a well-trained frozen GNN model $\varphi$ makes accurate predictions for samples close to the source distribution. Therefore, as training proceeds and $\varphi$ scores different random perturbations of the same samples, we utilize its confident predictions as pseudo-labels for optimization, as this helps $f$ to capture the distribution shift and make the predictions robust. We achieve this by passing $\mathcal{G}_p$ and $\mathcal{G}_w$ to $\varphi$ for prediction as:

$$\tilde{\mathbf{p}}_\varphi^{\mathcal{G}} = \delta(h(z_a^{\mathcal{G}}; \theta_h)) \qquad\qquad \hat{\mathbf{p}}_\varphi^{\mathcal{G}} = \delta(h(z_p^{\mathcal{G}}; \theta_h)) \qquad\qquad (1)$$

where $z_a^{\mathcal{G}} = g(\mathcal{G}_w; \theta_g)$ and $z_p^{\mathcal{G}} = g(\mathcal{G}_p; \theta_g)$, $\delta(.)$ denotes the softmax function, $\tilde{\mathbf{p}}_\varphi^{\mathcal{G}}, \hat{\mathbf{p}}_\varphi^{\mathcal{G}} \in \mathbb{R}^C$, and $C$ is the number of classes. Pseudo-labels are made as $\tilde{y}_\varphi^{\mathcal{G}} = \arg\max \tilde{\mathbf{p}}_\varphi^{\mathcal{G}}$, and finally, the consistency loss is:

$$\mathcal{L}_c = \frac{1}{|\mathcal{B}|} \sum_{\mathcal{G} \in \mathcal{B}} \mathbb{1}(\max(\tilde{\mathbf{p}}_\varphi^{\mathcal{G}}) > \tau) CE(\tilde{y}_\varphi^{\mathcal{G}}, \hat{\mathbf{p}}_\varphi^{\mathcal{G}}) \qquad\qquad (2)$$

where $CE(.,.)$ is the cross-entropy function, $\mathcal{B} = \{\mathcal{G}_i\}_{i=1}^{|\mathcal{B}|}$ is a sample batch, and $\tau$ is a confidence threshold. $\tau$ excludes low-certainty predictions (samples less aligned with the source data distribution) and can be fixed or class-dynamic (see Appendix A.5.3).

Unlike standard semi-supervised adaptation, UGPROMPT's use of consistency regularization is novel in that we apply it in a stricter frozen-GNN prompting setting, allowing only learning an external prompt that serves as a strong augmentation to align target samples with the source-trained model.

## 3.2   Prompt Regularization

**Diversity.** Due to the class imbalance, only reducing the consistency loss may cause biased predictions and trivial solutions such that every sample is assigned to the same class to reduce the overall entropy. To mitigate this, inspired by (Liang et al., 2020a), we regularize the model to maximize the entropy of the scores' expected value over a batch, and encourage diverse predictions.

$$\mathcal{L}_{div} = -H(\hat{\mathbf{q}}) = \mathbf{1}^\top (\hat{\mathbf{q}} \odot \log \hat{\mathbf{q}}); \ \hat{\mathbf{q}} = \frac{1}{|\mathcal{B}|} \sum_{\mathcal{G} \in \mathcal{B}} \hat{\mathbf{p}}_\varphi^{\mathcal{G}} \qquad\qquad (3)$$

Here $H(.)$ is the entropy function, and $\odot$ is the Hadamard product. Employing consistency regularization with an adjusted confidence threshold ($\tau$) and integrating a diversity loss ($\mathcal{L}_{div}$) to prevent class collapse helps our framework address pseudo-label dependency and potential calibration risks.

**Domain Adaptation.**   While the prompting function $f$ minimizes the discrepancy of predictions for the same sample, it may also create OOD prompted graphs. Since the source data are unavailable, we cannot directly align prompted representations with the source distribution. Instead, we can exploit the knowledge learned from source samples that is preserved in the frozen parameters $\theta_\varphi$ to mitigate the OOD issue. To achieve this, we train an adversarial discriminator $d(.; \theta_d)$—e.g., a simple feedforward network with trainable parameters $\theta_d$—to distinguish a prompted graph $\mathcal{G}_p$ from a non-prompted augmented graph $\mathcal{G}_w$. Formally, we optimize the discriminator as follows:

$$\theta_d^\star = \arg\min_{\theta_d} -\frac{1}{2|\mathcal{B}|} \sum_{\mathcal{G} \in \mathcal{B}} [\log \sigma(d(z_a^{\mathcal{G}}; \theta_d)) + \log(1 - \sigma(d(z_p^{\mathcal{G}}, \theta_d)))] \tag{4}$$

where $\sigma(.)$ is the sigmoid function. We normalize the sum by $\frac{1}{2|\mathcal{B}|}$ since every graph has two samples. Nevertheless, we regularize $f$ with the following objective to make $g$'s representations for prompted graphs closer to the non-prompted augmented graphs.

$$\mathcal{L}_{adv} = -\frac{1}{|\mathcal{B}|} \sum_{\mathcal{G} \in \mathcal{B}} \log \sigma(d(z_p^{\mathcal{G}}; \theta_d^\star)) \tag{5}$$

The adversarial discriminator regularizes the prompted representations to stay close to their non-prompted weak augmentations, preventing the prompt from moving them to arbitrary regions of the latent space. Thus, $L_{\text{adv}}$ should be interpreted as a target-side stability regularizer, while source compatibility is mainly induced by the frozen model and the confidence-filtered consistency objective. We provide an MMD-based diagnostic supporting this interpretation in Appendix A.8.

### 3.3 Final Objective & Complexity Analysis

Our unsupervised objective approach involves three parts. Eq. 2 encourages consistency across GNN predictions to handle distribution shift and exploit the GNN's learned knowledge from source data. Eq. 3 and Eq. 5 handle class imbalance and avoid generating OOD prompted graphs, respectively. The final objective to optimize $\theta_f$ becomes:

$$\theta_f^\star = \arg\min_{\theta_f} \mathcal{L}; \qquad \mathcal{L} = \mathcal{L}_c + \lambda_1 \mathcal{L}_{div} + \lambda_2 \mathcal{L}_{adv} \tag{6}$$

where $\lambda_1$, $\lambda_2$ are hyper-parameters. We bring empirical evidence supporting the logic behind adding each of the regularization objectives terms in Appendix A.7.2.

During inference, the augmentation step is skipped, allowing the prompting module to produce a prompted graph directly from an input graph to align it with the trained model's knowledge. This prompted graph is passed to the trained GNN for prediction. The pseudocode of UGPROMPT for both the training and inference is presented in Algorithms 1 and 2 of the Appendix.

The time complexity of a regular GNN (e.g. GCN), is $O(NLd^2 + L|E|d)$, where $N, E, L$, and $d$ are the number of nodes, edges, GNN layers, and the dimensionality of node embeddings respectively. A common graph augmentation algorithm, like feature masking, requires $O(Nd)$ operations. The complexity of the prompting method used in our experiments is $Ndn_t$, where $n_t$ is the number of trainable prompting vectors. Thus, the overall complexity is $O(NLd^2 + L|E|d + Ndn_t)$.

We compare wall-clock training and inference time against the baselines in Appendix A.7.6 (Table 21). UGPROMPT has slightly higher training cost due to consistency learning and the discriminator, but this overhead is only used during training. At inference time, the augmentation pipeline and discriminator are removed, and the model only applies the lightweight prompting function before the frozen GNN. Therefore, UGPROMPT remains close to the BaseModel inference cost while adding only a small number of trainable prompt parameters and a lightweight discriminator during training.

## 4 Experiments

**Datasets and Code.** We experiment on six standard datasets for graph and node classification. We use ENZYMES (Schomburg et al., 2004), PROTEINS (Borgwardt et al., 2005), DHFR (Sutherland et al., 2003), BBBP and BACE (Wu et al., 2017) datasets for graph classification, which have continuous or discrete

Table 1: Graph classification results on target datasets (for GCN and GAT base models) show our unsupervised UGPROMPT largely outperforming competitors that use 25% labeled data.

| Base GNN | Method | %Label | ENZYMES | | PROTEINS | | DHFR | | BBBP | | BACE | |
|---|---|---|---|---|---|---|---|---|---|---|---|---|
| | | | F1 | IMP | F1 | IMP | F1 | IMP | F1 | IMP | F1 | IMP |
| GCN | BaseModel | 0 | $47.7^{\pm5.7}$ | - | $51.8^{\pm7.0}$ | - | $75.5^{\pm6.3}$ | - | $87.3^{\pm1.2}$ | - | $58.5^{\pm3.3}$ | - |
| | Fine-Tuning | 25 | $46.4^{\pm0.1}$ -2.7 | | $47.5^{\pm0.7}$ -8.3 | | $76.8^{\pm0.1}$ 1.7 | | **$88.3^{\pm1.2}$** **1.1** | | $64.0^{\pm0.7}$ 1.1 | |
| | GraphPrompt | | $38.1^{\pm1.4}$ -20.1 | | $50.8^{\pm1.4}$ -1.9 | | $71.7^{\pm1.1}$ -5.0 | | $81.9^{\pm1.2}$ -6.2 | | $64.3^{\pm0.7}$ 9.9 | |
| | GraphPrompt+ | | $23.5^{\pm2.5}$ -50.7 | | $44.7^{\pm5.1}$ -13.7 | | $64.9^{\pm6.2}$ -14.0 | | $82.0^{\pm1.2}$ -6.1 | | $61.0^{\pm3.1}$ 8.4 | |
| | All-In-One | | $45.8^{\pm1.9}$ -4.0 | | $38.1^{\pm13.4}$ -26.4 | | **$79.1^{\pm0.6}$** **4.6** | | $85.7^{\pm1.2}$ -1.8 | | $52.5^{\pm6.1}$ -10.3 | |
| | GPF-Plus | | $48.3^{\pm1.8}$ 1.3 | | $53.8^{\pm2.6}$ 3.9 | | $76.9^{\pm0.3}$ 1.9 | | $88.0^{\pm1.2}$ 0.8 | | $63.4^{\pm1.2}$ 8.4 | |
| | UGPROMPT | **0** | **$49.1^{\pm0.6}$** **2.9** | | **$56.0^{\pm1.5}$** **8.1** | | $77.0^{\pm2.4}$ 2.0 | | **$88.3^{\pm0.2}$** **1.1** | | **$64.6^{\pm1.9}$** **10.4** | |
| GAT | BaseModel | 0 | $44.1^{\pm6.4}$ - | | $51.5^{\pm8.1}$ - | | $77.3^{\pm3.5}$ - | | $87.7^{\pm1.2}$ - | | $46.4^{\pm1.2}$ - | |
| | Fine-Tuning | 25 | $42.9^{\pm0.0}$ -2.7 | | $50.2^{\pm0.8}$ -2.5 | | $76.5^{\pm0.0}$ -1.0 | | **$88.4^{\pm0.1}$** **0.8** | | $46.4^{\pm0.7}$ -2.7 | |
| | GraphPrompt | | $29.3^{\pm1.0}$ -33.6 | | $49.5^{\pm1.1}$ -3.9 | | $72.8^{\pm0.8}$ -5.8 | | $83.4^{\pm0.3}$ -4.9 | | $57.9^{\pm0.5}$ 0.0 | |
| | GraphPrompt+ | | $25.7^{\pm4.6}$ -41.7 | | $48.7^{\pm7.7}$ -5.4 | | $57.2^{\pm9.1}$ -26.0 | | $82.4^{\pm2.0}$ -6.0 | | $64.2^{\pm3.7}$ 10.9 | |
| | All-In-One | | $39.5^{\pm2.5}$ -10.4 | | $31.5^{\pm14.8}$ -38.8 | | $76.0^{\pm2.0}$ -1.7 | | $86.5^{\pm0.4}$ -1.4 | | $57.9^{\pm3.9}$ -9.8 | |
| | GPF-Plus | | $42.9^{\pm2.0}$ -2.7 | | $54.9^{\pm1.9}$ 3.7 | | $77.3^{\pm0.9}$ 0.0 | | $88.3^{\pm0.3}$ 0.7 | | $65.8^{\pm0.9}$ 13.6 | |
| | UGPROMPT | **0** | **$45.9^{\pm2.2}$** **4.1** | | **$56.4^{\pm2.0}$** **9.5** | | **$78.2^{\pm0.9}$** **1.2** | | $87.8^{\pm0.2}$ 0.1 | | **$66.1^{\pm0.2}$** **14.2** | |

features. For node classification, we use Cora, CiteSeer, PubMed (Yang et al., 2016), Flickr (Zeng et al., 2020), Cornell, Texas, Wisconsin (Pei et al., 2020). Please find more details in Appendix A.5.1.

*Distribution Shift in Datasets.* Our problem definition (UGPP) requires evaluating a source-trained GNN on target datasets that exhibit a covariate shift from the source data. To implement this, our main experiments induce shifts based on fundamental graph properties. For graph classification, we generate datasets with varying edge homophily ratios (Zhu et al., 2020), a property known to intrinsically affect GNN information aggregation (Gilmer et al., 2017). For node classification, we use PageRank (Bazhenov et al., 2023) to create a popularity-based shift, which provides a challenging evaluation scenario. Further details on the generation of these distributions are in Appendix A.5.2, and additional experiments on other shifts (e.g., graph density, clustering coefficient) are in Appendix A.6.2.

**Evaluation Setting.** All datasets are split in half to create source and target sets with shifted distributions. We train a base GNN on the source set and evaluate it on the target set. Since baseline prompting methods rely on supervised training, we allow only the baselines to access labeled data for training in four setups of 25%, 50%, 75%, and 100% (full supervision). Please find experiments on 50%, 75% and 100% labeled data in Appendix A.9.1. We report the F1-score and the relative improvement in F1-score (IMP%) compared to the BaseModel baseline.

**Baselines.** We consider several types of baselines.
*(1) BaseModel.* The base GNN without prompting and fine-tuning, which is expected to be outperformed by prompting methods. We use GCN (Kipf & Welling, 2017) and GAT (Veličković et al., 2018) as the base GNN. More experiments with recent advanced GNNs are in Appendix A.6.3.
*(2) Fine-Tuning.* The base GNN model when we fix its encoder and just fine-tune its projection head. The goal is to verify the claim that fine-tuning on new dataset with labels does not necessarily improve performance (Sun et al., 2022; Fang et al., 2023; Sun et al., 2023).
*(3) GNN Prompting Methods.* Our work is the first attempt for graph prompting without labels and updating the base GNN's parameters. Thus, we compare with all the SOTA GNN prompting methods used in the recent benchmark (Zi et al., 2024), namely All-In-One (Sun et al., 2023) and GPF-Plus (Fang et al., 2023), GraphPrompt (Liu et al., 2023b), GraphPrompt+ (Yu et al., 2024a), and GPPT (Sun et al., 2022). Also, we have compared our method against additional baselines, using GCN as the BaseModel's architecture, in Tables 9 and 10 of the Appendix. We use the codebases from the corresponding papers.

We do not allow these methods to update the GNN's parameters and only their prompting modules are supposed to be learned on the target dataset. This restriction follows the UGPP setting, where the goal is to isolate the effect of the prompting function while keeping the entire source-trained model frozen. We note

Table 2: Node classification results on target datasets for GCN as the base model. Compared to baselines given 25% of labeled data, UGPROMPT generally achieves better results without labels.

| Base GNN | Method | %Label | Cora | | CiteSeer | | PubMed | | Flickr | | Cornell | | Texas | | Wisconsin | |
|---|---|---|---|---|---|---|---|---|---|---|---|---|---|---|---|---|
| | | | F1 | IMP | F1 | IMP | F1 | IMP | F1 | IMP | F1 | IMP | F1 | IMP | F1 | IMP |
| GCN | BaseModel | 0 | $53.8^{\pm2.4}$ | - | $44.1^{\pm1.5}$ | - | $57.1^{\pm0.8}$ | - | $\underline{16.5}^{\pm0.4}$ | - | $19.1^{\pm6.2}$ | - | $23.6^{\pm10.3}$ | - | $25.2^{\pm10.2}$ | - |
| | Fine-Tuning | 25 | $51.7^{\pm0.5}$ | -3.9 | $40.0^{\pm0.3}$ | -9.3 | $54.3^{\pm3.4}$ | -4.9 | $10.4^{\pm0.1}$ | -37.0 | $19.1^{\pm0.0}$ | 0.0 | $\underline{26.4}^{\pm0.0}$ | $\underline{11.9}$ | $\underline{27.1}^{\pm0.1}$ | $\underline{7.5}$ |
| | GPPT | | $47.8^{\pm3.5}$ | -11.2 | $38.4^{\pm0.5}$ | -12.9 | $51.6^{\pm4.8}$ | -9.6 | $13.5^{\pm0.5}$ | -18.2 | $15.1^{\pm3.0}$ | -20.9 | $25.6^{\pm8.6}$ | 8.5 | $23.4^{\pm0.1}$ | -7.1 |
| | GraphPrompt | | $53.8^{\pm0.4}$ | 0.0 | $41.6^{\pm0.3}$ | -5.7 | $56.9^{\pm0.1}$ | -0.4 | $13.0^{\pm0.1}$ | -21.2 | $10.9^{\pm1.4}$ | -42.9 | $4.8^{\pm0.4}$ | -79.7 | $10.5^{\pm0.1}$ | -58.3 |
| | GraphPrompt+ | | $49.8^{\pm0.3}$ | -7.4 | $39.9^{\pm0.3}$ | -9.5 | $\mathbf{62.0}^{\pm0.4}$ | $\mathbf{8.6}$ | $14.8^{\pm0.6}$ | -10.3 | $11.5^{\pm2.1}$ | -39.8 | $4.8^{\pm0.3}$ | -79.9 | $12.0^{\pm0.1}$ | -52.3 |
| | All-In-One | | $50.5^{\pm1.1}$ | -6.1 | $38.3^{\pm1.0}$ | -13.2 | $42.1^{\pm0.9}$ | -26.3 | $13.8^{\pm0.3}$ | -16.4 | $13.0^{\pm0.8}$ | -31.9 | $21.7^{\pm1.6}$ | -8.1 | $21.4^{\pm0.1}$ | -15.1 |
| | GPF-Plus | | $\underline{56.5}^{\pm0.6}$ | 5.0 | $\underline{45.6}^{\pm0.6}$ | 3.4 | $59.1^{\pm0.4}$ | 3.5 | $13.3^{\pm0.3}$ | -19.4 | $\underline{22.0}^{\pm0.5}$ | $\underline{15.2}$ | $25.2^{\pm1.4}$ | 6.8 | $26.7^{\pm0.1}$ | 6.0 |
| | UGPROMPT | 0 | $\mathbf{57.3}^{\pm0.4}$ | $\mathbf{6.5}$ | $\mathbf{45.7}^{\pm0.4}$ | $\mathbf{3.6}$ | $\underline{61.2}^{\pm0.3}$ | $\underline{7.2}$ | $\mathbf{17.5}^{\pm0.4}$ | $\mathbf{6.1}$ | $\mathbf{23.2}^{\pm0.1}$ | $\mathbf{21.5}$ | $\mathbf{26.8}^{\pm0.8}$ | $\mathbf{13.6}$ | $\mathbf{28.0}^{\pm0.1}$ | $\mathbf{11.1}$ |
| GAT | BaseModel | 0 | $\underline{47.7}^{\pm1.3}$ | - | $41.2^{\pm2.4}$ | - | $60.0^{\pm1.1}$ | - | $17.0^{\pm0.2}$ | - | $\underline{18.6}^{\pm0.2}$ | - | $28.1^{\pm0.2}$ | - | $19.9^{\pm6.9}$ | - |
| | Fine-Tuning | 25 | $43.5^{\pm0.6}$ | -8.8 | $38.8^{\pm0.3}$ | -5.8 | $55.6^{\pm2.7}$ | -7.3 | $10.9^{\pm0.2}$ | -35.9 | $18.2^{\pm0.0}$ | -2.2 | $21.2^{\pm0.0}$ | -24.6 | $\underline{21.8}^{\pm0.0}$ | $\underline{9.5}$ |
| | GPPT | | $31.5^{\pm3.9}$ | -34.0 | $34.3^{\pm1.8}$ | -16.7 | $51.7^{\pm4.6}$ | -13.8 | $12.9^{\pm0.1}$ | -24.1 | $17.2^{\pm4.5}$ | -7.5 | $28.2^{\pm5.5}$ | 0.4 | $21.5^{\pm4.1}$ | 8.0 |
| | GraphPrompt | | $44.2^{\pm0.6}$ | -7.3 | $39.2^{\pm0.4}$ | -4.9 | $60.1^{\pm0.1}$ | 0.2 | $13.4^{\pm0.3}$ | -21.1 | $14.3^{\pm1.3}$ | -23.1 | $1.4^{\pm0.0}$ | -95.0 | $15.4^{\pm1.5}$ | -22.6 |
| | GraphPrompt+ | | $41.2^{\pm0.9}$ | -13.6 | $37.8^{\pm0.7}$ | -8.3 | $\mathbf{64.0}^{\pm1.1}$ | $\mathbf{6.7}$ | $\underline{17.5}^{\pm0.6}$ | $\underline{2.9}$ | $13.5^{\pm2.0}$ | -27.4 | $1.4^{\pm0.0}$ | -95.0 | $17.1^{\pm2.4}$ | -14.1 |
| | All-In-One | | $34.3^{\pm2.1}$ | -28.1 | $27.6^{\pm1.4}$ | -33.0 | $22.7^{\pm3.2}$ | -62.2 | $13.3^{\pm0.2}$ | -21.8 | $13.5^{\pm0.2}$ | -27.4 | $21.2^{\pm0.7}$ | -24.6 | $16.9^{\pm0.9}$ | -15.1 |
| | GPF-Plus | | $47.6^{\pm1.5}$ | -0.2 | $\underline{42.1}^{\pm0.6}$ | $\underline{2.2}$ | $60.1^{\pm0.3}$ | 0.2 | $13.8^{\pm0.2}$ | -18.8 | $17.9^{\pm1.1}$ | -3.8 | $\mathbf{30.4}^{\pm0.7}$ | $\mathbf{8.2}$ | $21.7^{\pm1.2}$ | 9.0 |
| | UGPROMPT | 0 | $\mathbf{48.8}^{\pm0.9}$ | $\mathbf{2.3}$ | $\mathbf{42.3}^{\pm0.5}$ | $\mathbf{2.7}$ | $\underline{60.2}^{\pm0.1}$ | $\underline{0.3}$ | $\mathbf{17.6}^{\pm0.3}$ | $\mathbf{3.5}$ | $\mathbf{21.8}^{\pm1.5}$ | $\mathbf{17.2}$ | $\underline{29.5}^{\pm1.0}$ | $\underline{5.0}$ | $\mathbf{22.2}^{\pm1.1}$ | $\mathbf{11.6}$ |

that this is more restrictive than the original setting of the supervised prompting baselines, which often fine-tune a task-specific prediction head. Therefore, in Appendix A.7, we additionally evaluate these baselines under a relaxed setting where their decoders are fine-tuned with labeled target data.

## 4.1 Results on Graph Classification

For graph classification, 50% of graphs are randomly sampled as the source dataset, with higher-homophily graphs having a greater chance of selection; the remaining 50% form the target dataset. Please see experiments with graph density distribution shift in Table 11 (see the Appendix). The base GNN is trained on the source dataset. We repeat the experiments with two base GNNs (GCN and GAT) to demonstrate how the models generalize across different architectures. Note that GPPT is limited to node classification, so we exclude it from this experiment.

Table 1 presents graph classification results, where baselines use 25% labeled data, while our method, UG-PROMPT, uses 0%. Two key observations highlight UGPROMPT's contribution: first, it consistently surpasses the BaseModel, validating it as a reliable, non-detrimental prompting method. Second, and *most notably, UGPROMPT's use of no labels offers broad applicability to diverse unlabeled datasets, marking a step towards graph foundation models.* Interestingly, evaluations with UGPP setting reveal that baselines often fail to improve performance and sometimes make it worse. Most of the graph prompting methods, except for GPF-Plus, perform poorly with both GNN architectures. Although GPF-Plus has the same prompting function as UGPROMPT's, it struggles with adapting to distribution shifts. Conversely, UGPROMPT leverages source data knowledge and generates pseudo-labels from highly confident predictions, learning effectively from samples that closely match the source distribution. This ensures consistent improvement across all cases.

## 4.2 Results on Node Classification

First, we compute PageRank (PR) for all nodes of each dataset. We sample 50% of nodes from the source dataset based on the normalized PR, so that graphs with higher PR are more likely to be included. The rest are assigned to the target dataset. A 2-hop neighborhood of each node is extracted as a subgraph for task unification to graph classification, and it inherits the main node's label. More experiments with the distribution shift of the type clustering coefficient are provided in Table 12 (see the Appendix).

Node classification results are in Table 2. UGPROMPT outperforms on all datasets except PubMed (where it is second-best) and, *notably, uses no labeled data, unlike the baselines.* Additionally, the GNN's performance

Table 3: UGPROMPT performance under different strong augmentation rates $p_s$ for the prompted augmented graph. Higher values of $p_s$ provide greater variation in masked feature groups and generally help better capture distribution shifts.

| $p_s$ | ENZYMES | | PROTEINS | | DHFR | | Cora | | CiteSeer | | PubMed | |
|---|---|---|---|---|---|---|---|---|---|---|---|---|
| | F1 | IMP | F1 | IMP | F1 | IMP | F1 | IMP | F1 | IMP | F1 | IMP |
| 0.0 | 47.7 | - | 55.7 | - | 76.8 | - | 56.3 | - | 44.9 | - | 57.2 | - |
| 0.1 | **49.1** | **2.9** | 55.7 | 0.0 | **77.0** | **0.3** | 56.7 | 0.7 | 44.9 | 0.0 | 58.6 | 2.4 |
| 0.2 | 48.9 | 2.5 | 55.9 | 0.4 | 76.8 | 0.0 | 56.8 | 0.9 | 45.0 | 0.2 | 59.7 | 4.4 |
| 0.3 | 48.1 | 0.8 | **56.0** | **0.5** | 76.2 | -0.8 | 57.1 | 1.4 | 45.2 | 0.7 | 60.4 | 5.6 |
| 0.4 | 46.9 | -1.7 | 55.6 | -0.2 | 75.7 | -1.4 | **57.3** | **1.8** | **45.7** | **1.8** | **61.2** | **7.0** |

degrades on target data across all datasets after fine-tuning its projection head (the Fine-Tuning baseline) with 25% of labels. This verifies that fine-tuning a model on a small-sized labeled dataset may introduce noisy information when the downstream data distribution does not align with the model's learned knowledge (Bousquet et al., 2004; Shalev-Shwartz & Ben-David, 2014; Brown et al., 2020). We also verify that UGPROMPT maintains high performance on the source domain and does not cause forgetting of learned knowledge, unlike the other baselines; please find the experiments in Appendix A.7.3.

An important advantage of an unsupervised method—such as ours—is that it allows the use of large-scale unlabeled datasets. It is important to emphasize that UGPROMPT achieves the best results on Flickr, the largest dataset, and second-best with large margins over other baselines on PubMed, the second-largest dataset, indicating that UGPROMPT can perform well on large datasets.

Although supervised learning usually provides more informative gradients, this is not necessarily an upper bound in the UGPROMPT setting. Since the source-trained GNN is frozen and the target data are shifted, adaptation mainly requires aligning target samples with the source-trained model. UGPROMPT uses confidence-thresholded pseudo-labeling and therefore optimizes the prompt only from target samples on which the source-trained model is already certain, reducing the impact of highly shifted samples. Moreover, UGPROMPT allows the use of the full unlabeled target set, whereas supervised baselines are limited to training on the labeled subset. The few-shot results in Section 4.4 further show that labels improve UGPROMPT when they are used together with our prompt-based alignment.

Additionally, we evaluate the supervised prompting baselines in their more relaxed setting with fine-tuned decoders and 25% labeled target data in Appendix A.7; UGPROMPT remains more stable under distribution shift despite using no labels and keeping the full base model frozen.

## 4.3 Ablation Study

### 4.3.1 The Effect of Regularization

To evaluate the regularization effect, UGPROMPT is trained in four scenarios: (1) without regularization ("w/o"), (2) with only domain adaptation regularization ("domain"), (3) with only diversity regularization ("diversity"), and (4) with both regularizations ("domain + diversity"). Settings (2), (3), and (4) are compared to (1), with Figure 2 showing IMP improvements. During evaluation, if validation performance indicates that a specific regularization term fails to improve

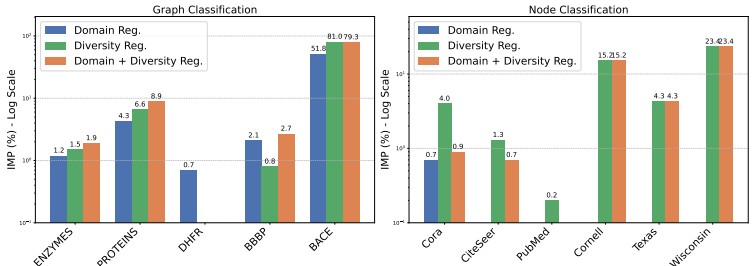

Figure 2: The effect of regularization objectives on UGPROMPT with GCN as the base model.

the model, its corresponding weight ($\lambda_1$ or $\lambda_2$ in Equation 6) is set to zero, and the test IMP for that term resolves to zero. Results show that both regularization factors have positive effects across most datasets; however, their combination is not always superior to their individual applications.

Table 4: Scaling with homophily shift across four target sets. UGPROMPT is better in most cases.

| Target set | Method | ENZYMES | PROTEINS | DHFR | BACE | Cora | CiteSeer |
|---|---|---|---|---|---|---|---|
| Set 1 | GPF-Plus | -3.6 | 11.5 | -3.2 | 20.7 | -1.8 | 0.2 |
| | UGPrompt | **0.9** | **19.6** | **-0.6** | **40.1** | **0.7** | **2.0** |
| Set 2 | GPF-Plus | -5.4 | 57.6 | -1.9 | 106.1 | 2.4 | 0.2 |
| | UGPrompt | **1.0** | **62.0** | **2.0** | **155.6** | **2.6** | **0.7** |
| Set 3 | GPF-Plus | **27.1** | 85.7 | **10.2** | 108.8 | **2.2** | **2.1** |
| | UGPrompt | 3.0 | **89.2** | 7.3 | **124.7** | 0.2 | 0.5 |
| Set 4 | GPF-Plus | -7.8 | 44.4 | **3.7** | 97.3 | -2.6 | 1.3 |
| | UGPrompt | **1.9** | **46.4** | 2.7 | **113.7** | **3.5** | **1.9** |

Domain adaptation regularization is beneficial across all graph classification datasets. We conjecture that the distributions of node classification datasets are more likely to be densely populated, whereas graph datasets often exhibit scattered hollow spaces in the latent space between classes, increasing the likelihood of generating OOD-prompted graphs. Therefore, domain adaptation regularization would be more beneficial. Moreover, a key finding of this experiment is the importance of adding diversity regularization. More specifically, when we have severe class imbalance, for example, in PROTEINS, Cora, and CiteSeer, "diversity" empowers the base GNN significantly, which supports our claims. Appendix A.7.2 provides more discussion on the effect of these objectives.

### 4.3.2 The Effect of Augmentation

We mask node features in weakly augmented graphs $(\mathcal{G}_w)$ for pseudo-labeling with probability $p_w$ and in strongly augmented graphs $\mathcal{G}_s$ for prompting with probability $p_s$. Since $\mathcal{G}_s$ resembles a distribution shift from $\mathcal{G}_w$, a higher $p_s$ enables learning from distribution shifts across more feature groups, potentially offering greater robustness. Here, we fix $p_w = 0.1$ and evaluate the impact of varying $p_s$. The IMP for all $p_s$ values is compared to no augmentation $p_s = 0$. The results in Table 3 support our intuition that augmenting, confidence pseudo-labeling, and consistency training are advantageous. Higher $p_s$ yields better performance over PROTEINS, Cora, CireSeer, and PubMed because augmentation by feature masking—replacing a group of feature values with 0.0—is semantically more aligned with the discrete and binary features of these datasets; however, for ENZYMES and DHFR with continuous features, this augmentation is not as effective and higher values of $p_s$ may be detrimental and cause learning from noise. Additional experiments on evaluating the effect of augmentation type (e.g. modifying graph structure) are in Appendix A.7.7. Besides, we show the versatility and generalization of our framework in Appendix A.7.6 using All-In-One's prompting function as a choice of function $f$ (see Section 3.1). Also, additional diagnostics on pseudo-label accuracy, calibration, and sensitivity to confidence threshold $(\tau)$ are provided in Appendix A.9.

### 4.3.3 The Effect of Homophily Shift

To evaluate how performance scales with the increase in covariate shift (or difficulty), we extend our experiment by constructing explicit degrees of shift. Instead of a random 50–50 split (as in the original setting), we sort all samples by homophily (for the graph classification dataset) and PageRank (for node classification datasets). The top 50% (highest homophily) are used as the source dataset. The remaining 50% are divided into four target sets with increasing difficulty (amount of covariate shift with respect to source set): (i) Set 1: 50–67.5 (closest to source), (ii) Set 2: 67.5–75, (iii) Set 3: 75–87.5, and (iv) Set 4: 87.5–100 (lowest homophily). Table 4 shows the results. UGPROMPT maintains strong gains on the easier shifts (Sets 1 & 2) and continues to deliver improvements even under severe distribution shift (Sets 3 & 4). In contrast, GPF-Plus is worse and frequently induces negative transfer (negative IMP%).

### 4.4 Few-shot Learning for UGPrompt

UGPROMPT is an unsupervised GNN prompting method. However, it can potentially utilize labeled data efficiently when it is available. Assuming every batch $\mathcal{B}$ is composed of a set of labeled samples $S_l$ and unlabeled samples $S_u$ s.t. $\mathcal{B} = S_l \cup S_u$, we can replace $\mathcal{L}_c$ in Equation 2 by $\mathcal{L}_c = \frac{1}{|\mathcal{B}|}(\mathcal{L}_l + \lambda_3 \mathcal{L}_u)$ in which

Table 5: Evaluation of UGPROMPT 's performance in a few-shot setting using GCN as the base GNN under homophily and PR distribution shifts. Results indicate considerable gains with labeled data, particularly in node classification tasks.

| Method | %Label | ENZYMES | | PROTEINS | | DHFR | | Cora | | CiteSeer | | PubMed | |
|---|---|---|---|---|---|---|---|---|---|---|---|---|---|
| | | F1 | IMP | F1 | IMP | F1 | IMP | F1 | IMP | F1 | IMP | F1 | IMP |
| BaseModel | | 47.7 | - | 51.8 | - | 75.5 | - | 53.8 | - | 44.1 | - | 57.1 | - |
| UGPROMPT (ours) | 25 | **49.1** | **2.9** | **56.2** | **8.5** | **77.2** | **2.3** | **58.2** | **8.2** | **47.0** | **6.6** | **63.5** | **11.2** |
| | 10 | _49.0_ | _2.7_ | _56.1_ | _8.3_ | 77.0 | 2.0 | _57.6_ | _7.1_ | _46.2_ | _4.8_ | _62.1_ | _8.8_ |
| | 5 | **49.1** | **2.9** | _56.1_ | _8.3_ | _77.1_ | _2.1_ | 57.5 | 6.9 | 46.0 | 4.3 | _62.1_ | _8.8_ |
| | 0 | **49.1** | **2.9** | 56.0 | 8.1 | 77.0 | 2.0 | 57.3 | 6.5 | 45.7 | 3.6 | 61.2 | 7.2 |

Table 6: Category-disjoint stress test between source and target datasets, while the projection head remains frozen. Upper panel: BaseModel F1-score on source and target domains. Lower panel: prompting methods' improvements (IMP%) on the target domain.

| Method | Cornell | Texas | Wisconsin |
|---|---|---|---|
| BaseModel Performance (F1-score) | | | |
| Source | 46.7 | 79.8 | 60.1 |
| Target | 15.3 | 13.1 | 13.9 |
| Target Domain Improvement (IMP%) | | | |
| GPF-Plus | 30.1 | -1.5 | 5.4 |
| **UGPrompt** | **36.6** | **9.9** | **28.1** |

$\mathcal{L}_l = \sum_{\mathcal{G} \in S_l} CE(y^{\mathcal{G}}, \hat{\mathbf{p}}_{\varphi}^{\mathcal{G}})$ is the supervised objective term and $\mathcal{L}_u = \sum_{\mathcal{G} \in S_u} \mathbb{1}(\max(\tilde{\mathbf{p}}_{\varphi}^{\mathcal{G}}) \geq \tau) CE(\tilde{y}_{\varphi}^{\mathcal{G}}, \hat{\mathbf{p}}_{\varphi}^{\mathcal{G}})$ is the unsupervised term.

Here we evaluate UGPROMPT in a few-shot setting. Table 5 presents the results using GCN as the base GNN under homophily and PR distribution shifts. UGPROMPT significantly performs better when labels are provided. For node classification datasets, IMP in the 25% labels setting is notably higher than in the unsupervised case (0% labels), while improvements on PROTEINS and DHFR are smaller. Meanwhile, performance on ENZYMES remains comparable in the absence of labels, likely due to a significant covariate shift, which makes learning from highly heterophilic data challenging even with labels. The key takeaway emerges by comparing Table 5 with Tables 1 and 2, showing that with 25% of labeled data, UGPROMPT outperforms all baselines on most datasets except DHFR. Notably, its superiority is also evident in the 0% label setting.

## 4.5 Exploratory Robustness Beyond the UGPP Assumption

The formal UGPP setting assumes covariate shift with a shared conditional label distribution. Here, we provide an exploratory stress test beyond this assumption. The goal is to evaluate whether prompting can still improve a frozen model when the target categories are semantically disjoint from those used to train the head. We use the Cornell, Texas, and Wisconsin datasets, which share the same feature dimensionality and number of classes. For each dataset, the base GNN is trained only on source samples from classes $\{0, 1, 2\}$, and then the base model is frozen. We then prompt and evaluate it on samples from classes $\{2, 3, 4\}$. Since the projection head remains frozen with output dimension $C = 3$, the target labels are reindexed to $\{0, 1, 2\}$ only for computing evaluation metrics, and no classifier weights are updated.

Table 6 summarizes the findings. The BaseModel collapses under this category-disjoint stress test, showing that the original semantic mapping of the frozen head is no longer reliable. Nevertheless, UGPROMPT recovers substantial performance across all datasets, outperforming GPF-Plus, which occasionally leads to negative transfer.

Our second experiment is designed to evaluate robustness to real-world covariate shift. We consider a more extreme scenario than the simulated covariate shift discussed before. This setting is standard (particularly in transfer learning) and more challenging. Here, we train the base GNN on a source dataset, such as

Table 7: Cross-dataset transfer results. All values are IMP%. UGPROMPT consistently delivers positive transfer. C, T, and W denote Cornell, Texas, and Wisconsin, respectively.

| | C→T | C→W | T→C | T→W | W→C | W→T |
|---|---|---|---|---|---|---|
| GPF-Plus | -3.4 | 81.1 | -5.0 | 61.9 | 33.0 | 209.8 |
| **UGPrompt** | 11.1 | 10.4 | 6.2 | 1.2 | 41.5 | 123.5 |

Cornell, and then prompt and evaluate it on another dataset, such as Texas, which is denoted as Cornell → Texas. Results are shown in Table 7. UGPROMPT consistently delivers positive transfer, whereas GPF-Plus potentially produces negative transfer. This demonstrates UGPROMPT's robustness even under severe shifts.

## 5 Conclusions

In conclusion, we have introduced UGPROMPT, a novel unsupervised prompting framework for GNNs that overcomes the limitations of existing prompting methods, particularly in scenarios where labeled data is unavailable. UGPROMPT eliminates the need for updating the base GNN's parameters on new downstream tasks. UGPROMPT also enhances the generalization of the base source-trained GNNs without supervision. Experimental results over various datasets validate the effectiveness of UGPROMPT which outperforms the state-of-the-art prompting methods that rely on labeled data on both graph and node classification tasks in many settings.

*Limitations & Future Work.* Since we do not involve training the projection head of the source-trained GNN for downstream tasks, UGPROMPT assumes that the source and target domains share the same label space. Thus, it is not designed for a general label distribution shift with new target classes. This limitation is tied to UGPROMPT's frozen-head requirement. The lack of a unified discrete output space across graph tasks and datasets requires retraining or fine-tuning a task-specific projection head to handle new labels, which falls outside our UGPP setting. Developing universal graph vocabularies or output spaces for open-vocabulary graph prompting is an important future direction. Another interesting future direction would be to develop a method that selects high-quality pseudolabels in the presence of a severe covariate shift.

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

# A Appendix

## A.1 Reproducibility.

We have made the codebase and other implementation details available in this anonymous link: `https://anonymous.4open.science/r/UGPrompt-6C3E`

## A.2 Additional Related Work

**Prompting for LLMs.** The promising results of early LLMs (Radford & Narasimhan, 2018; Devlin et al., 2019) have inspired prompt-tuning approaches that leverage LLMs' reasoning capabilities with minimal parameter tuning. These modular methods (Li & Liang, 2021; Lester et al., 2021) integrate trainable prompting prefixes with LLMs, with their parameters frozen, offering remarkable performance across various tasks while reducing complexity. Besides, (Radford et al., 2019) suggested pre-training on a large, diverse corpus (WebText) and demonstrated strong zero-shot performance across various tasks. Following that, (Brown et al., 2020) coined the term in-context learning, which refers to the effort to help models, particularly LLMs, generalize to new tasks without any parameter training, only by constructing informative prompts with task descriptions, examples, and instructions. The approaches propose specific language hints to guide the reasoning process, for example, by providing fine-grained and conditionally specific instructions (Mishra et al., 2022), or by encouraging the LLM to engage in sequential reasoning with (Wei et al., 2022) or without (Kojima et al., 2022) examples. Additionally, some methods iteratively score, evaluate, and update prompts in refinement loops (Zhou et al., 2023; Yang et al., 2024a).

**Prompting for GNNs.** A few studies have also adopted prompt tuning for GNNs. Specifically, the main track of these works starts with the "pre-train, prompt, fine-tune" paradigm proposed by (Sun et al., 2022); they address the common issue of a discrepancy between training and test objectives that causes performance drop in downstream applications. They design a pre-training task, specifically edge prediction, that can align with the downstream task. Nonetheless, their approach is only applicable to node classification. Later, GraphPrompt (Liu et al., 2023b) and GraphPrompt+ (Yu et al., 2024a) propose subgraph similarity detection as a more general *task unification* for different graph tasks to pre-train a GNN encoder. Their prompting method involves task-specific trainable readout functions. Similarly, PRODIGY (Huang et al., 2023) generates prompts as a combination of example subgraphs, connected to label nodes, and query subgraphs, awaiting connection to label nodes, and pre-trains with a neighborhood-matching task. Unification to subgraph classification is also proposed by GPF-Plus (Fang et al., 2023) and ALL-In-One (Sun et al., 2023). However, these methods resemble LLM prompting methods more, in that they prompt input graphs before a frozen GNN encoder. Specifically, the former adds trainable prompting parameters to the feature matrix of an input graph, and the latter combines a subgraph with a trainable feature matrix and structure with the input. More recently, GCOPE (Zhao et al., 2024) utilizes graph coordinators to align domain-shifted datasets and remedy negative transfer in cross-domain pre-training. Also, DAGPrompt (Chen et al., 2025) introduces distribution-aware prompting by utilizing low-rank adaptation for heterophilic graphs. Similarly, to capture diverse node-specific patterns in non-homophilic graphs, PRONOG (Yu et al., 2025b) introduces a conditional prompting. In addition, Prompting GNNs for dynamic graphs has also been studied lately (Yu et al., 2025a).

There are two shortcomings with the above GNN prompting methods. First, all of them require labeled data for training their prompting functions or for testing. Second, they mostly train a new projection head/decoder for the pre-trained GNN along with the prompting parameters. However, we propose a fully unsupervised prompting method that does not require fine-tuning and achieves promising performance even when competitors have access to all or a subset of the labels.

**Consistency Regularization and Pseudo-labeling.** In the context of Semi-Supervised Learning (SSL), pseudo-labeling (Lee, 2013; Xie et al., 2020) and consistency regularization (Bachman et al., 2014; Sajjadi et al., 2016; Laine & Aila, 2017) for training neural network models when labeled data are scarce. The first technique augments the labeled dataset with the model's predictions on the unlabeled dataset, and the second aims to minimize the discrepancy in a model's predictions across random perturbations of the same

samples generated by augmentation and dropout layers. In particular, these methods have been applied to domain adaptation to mitigate distribution shifts between source and target datasets (Liang et al., 2020b; Kim et al., 2021). A widely studied approach to integrating pseudo-labeling and consistency regularization is to first generate random weakly and strongly augmented instances from the same dataset samples, and then assign pseudo-labels to the weakly augmented samples whenever the model makes confident predictions for them. Models are trained by these pseudo-labels as labels for the strongly augmented samples, along with labeled data if available. To select confident samples, some methods use a fixed certainty threshold (Sohn et al., 2020) while others set dynamic class-wise thresholds (Zhang et al., 2021; Wang et al., 2023).

Unlike previous work that used consistency regularization and pseudo-labeling, we are not interested in training or fine-tuning a model. Therefore, our novelty lies in interpreting a prompted graph as a strongly augmented instance and using the pseudo-labels from weak augmentation to train the prompting parameters.

### A.3 Algorithm

Algorithm 1 presents our method's prompting procedure during training. In line 1, we initialize the base GNN model with trained parameters tuned on the source and target datasets, the augmentation and prompting functions, and the method's hyperparameters. We initialize the prompting parameter and fix the base GNN's parameters in lines 3-4. Line 6 shows sampling of a batch of graphs and in line 7 a strong and a weak augmentation is generated from each graph in the batch. Next, the strongly augmented graph is prompted in line 8. This prompted graph, along with the weakly augmented graph, is encoded by GNNs, as in line 9. In line 10, the representation of the prompted and weakly augmented graphs is used to optimize the discriminator. Finally, in lines 9-10, the GNN's projection head (decoder) decodes the representation, and the diversity, domain adaptation, and consistency objective functions are used to optimize the prompting parameters. Algorithm 2 shows the inference stage of our method. At inference time, a test sample is passed directly to the prompting function without augmentation, and the GNN scores the prompted graph.

---

**Algorithm 1** UGPROMPT (Training)

---

1: **Input:** Target unlabeled dataset $\mathcal{D} = \{\mathcal{G}_i\}_{i=1}^{N_t}$, confidence threshold $\tau$, GNN model $\varphi = h(.; \theta_h) \circ g(.; \theta_g)$, number of trainable prompting parameters $n_p$, augmentation function $aug(.; p)$, weak augmentation probability $p_w$, and strong augmentation probability $p_s$.

2: **Output:** Prompting function $f(.; \theta_f)$ with optimized parameters $\theta^\star$.

3: Initialize prompting parameters $\theta_f$.
4: Freeze the parameters $\theta_g$ and $\theta_h$ of the base GNN.
5: **while** not converged **do**
6:     Sample batch of graphs $\mathcal{B} = \{\mathcal{G}_i\}_{i=1}^{|\mathcal{B}|} \subset \mathcal{D}$.
7:     For every graph $\mathcal{G} \in \mathcal{B}$ make a weak augmentation $\mathcal{G}_w \leftarrow aug(\mathcal{G}, p_w)$ and a strong augmentation $\mathcal{G}_s \leftarrow aug(\mathcal{G}, p_s)$.
8:     Prompt every strongly augmented graph as $\mathcal{G}_p \leftarrow f(\mathcal{G}_s; \theta_f)$.
9:     Using encoder $g$, encode every weakly augmented graph as $z_a^{\mathcal{G}} = g(\mathcal{G}_w; \theta_g)$ and every prompted graph as $z_p^{\mathcal{G}} = g(\mathcal{G}_p; \theta_g)$.
10:    Pass all encodings $z_a^{\mathcal{G}}$ and $z_p^{\mathcal{G}}$ to domain discriminator and optimize its parameters $\theta_d$ by Equation (4).

11:    Using decoder $h$, compute prediction scores for all encodings $z_a^{\mathcal{G}}$ and $z_p^{\mathcal{G}}$ as in Equation (1).
12:    Optimize for the prompting parameters $\theta_f$ by Equation (6).
13: **end while**
14: **return** $f(.; \theta^\star)$.

---

### A.4 Remark on UGPP definition.

Firstly, our problem setting differs from out-of-distribution (OOD) generalization methods (Li et al., 2023; Fan et al., 2024; Yang et al., 2024b). In OOD generalization, given samples of dataset $\mathcal{D} = \{(x_i, y_i)\}_{i=1}^{N}$

---

**Algorithm 2** UGPROMPT (Inference)

---

1: **Input:** Target unlabeled dataset $\mathcal{D} = \{\mathcal{G}_i\}_{i=1}^{N_t}$, confidence threshold $\tau$, freezed pretrained GNN $\varphi = h(.;\theta_h) \circ g(.;\theta_g)$, optimized prompting function $f(.;\theta_f^\star)$
2: **Output:** Prediction score set $\text{PRED}_\mathcal{D} = \{\varphi(f(\mathcal{G}_i;\theta_f^\star);\theta_h,\theta_g); \mathcal{G}_i \in \mathcal{D}\}_{i=1}^{N_t}$

3: Initialize empty set $\text{PRED} = \{\}$.
4: **for** $\mathcal{G} \in \mathcal{D}$ **do**
5:     Directly pass $\mathcal{G}$ to $f$ and make a prompted graph as $\mathcal{G}_p \leftarrow f(\mathcal{G};\theta_f^\star)$.
6:     Label the prompted graph using the source-trained GNN as $\hat{y}_\varphi^\mathcal{G} = \arg\max \varphi(\mathcal{G}_p;\theta_h,\theta_g)$
7:     Add $\mathcal{G}$'s label to the to the prediction set as $\text{PRED} \leftarrow \text{PRED} \cup \{\hat{y}_\varphi^\mathcal{G}\}$
8: **end for**
9: **return** PRED.

---

drawn from training distribution $P_{train}(X,Y)$, the goal is to train an optimal model $f(.;\theta)$ to have the best generalization to the test samples drawn from the distribution $P_{test}(X,Y)$, where $P_{train}(X,Y) \neq P_{test}(X,Y)$. This differs from our problem setting, as our goal is to propose a prompting method for GNNs that aligns with the in-context learning paradigm of LLMs. As discussed, LLM prompting methods commonly 1) do not retrain or fine-tune the LLM, 2) do not necessarily use labeled data, and 3) do not assume access to the data used for training the LLM. UGPP directly encourages these properties. We assume the GNN $\varphi(.;\theta_g,\theta_h)$ is first trained on distribution $\mathcal{P}_X^s \mathcal{P}_{Y|X}^s$ and $\mathcal{D}_s$ has the same train and test distributions, i.e. $\mathcal{P}_{X,Y}^{s_{train}} = \mathcal{P}_{X,Y}^{s_{test}} = \mathcal{P}_X^s \mathcal{P}_{Y|X}^s$, while this model is aimed to be used on $\mathcal{D}_t$ with unlabeled training distribution $\mathcal{P}_{X,Y}^{t_{train}}$ and test distribution $\mathcal{P}_{X,Y}^{t_{test}}$, such that $\mathcal{P}_X^{t_{train}} = \mathcal{P}_X^{t_{test}} \neq \mathcal{P}_X^s$, but $\mathcal{P}_{Y|X}^{t_{train}} = \mathcal{P}_{Y|X}^{t_{test}} = \mathcal{P}_{Y|X}^s$ and $\mathcal{P}_{Y|X}^{t_{train}}$ is not available. So far, this assumption of UGPP makes it close to the unsupervised domain adaptation (UDA) problem (You et al., 2019; Farahani et al., 2021) and satisfies unsupervised learning on $\mathcal{D}_t$, the second property of LLM in-context learning methods discussed above. For the third property, we assume the source dataset $\mathcal{D}_s$ is unobservable after training, which is known as source-free domain adaptation (SFDA) (Yang et al., 2021; Li et al., 2024). For the first property, we assume all the trained GNN's parameters $(\theta_g, \theta_h)$ are fixed and we do not fine-tune them on dataset $\mathcal{D}_t$, analogous to LLM in-context learning. So, our work also differs from SFDA methods, which allow fine-tuning all or a portion of the parameters on $\mathcal{D}_t$. In summary, our problem setting differs from UDA, SFDA, and OOD generalization, and it encourages the evaluation and design of generalized prompting methods for GNNs.

### A.5 Additional Details on Experimental Setup

### A.5.1 Datasets Statistics

In this work, we use datasets with different tasks and types of features. For graph classification, we evaluate on bioinformatics and molecular datasets, specifically ENZYMES (Schomburg et al., 2004) for multi-class classification and DHFR (Sutherland et al., 2003) for binary classification (both with continuous features), PROTEINS (Borgwardt et al., 2005) with discrete features for binary classification, also on BBBP and BACE (Wu et al., 2017). For node classification, we use common citation networks Cora, CiteSeer, PubMed (Yang et al., 2016), and Flickr (Zeng et al., 2020) with discrete features of online images for multi-class classification. We also experimented on Cornell, Texas, and Wisconsin (Pei et al., 2020) as semantic web datasets with discrete features. Table 8 shows the statistics of these datasets.

### A.5.2 Details on the Distribution Shift

Introducing distribution shift to graphs is challenging. First, the position of nodes does not matter in graphs, therefore, introducing distribution shift by geometric transformations is not an option. Second, it is hard to find invariant features among all graphs for universal manipulations to inject shift—like color manipulations in image domain since color channels are common features. Moreover, random perturbation to node features cannot be seen as a distribution shift because it may lead to noisy datasets rather than a distribution shift

Table 8: Statistic of the datasets used for experiments.

| Dataset | #Classes | #Graphs | #Nodes | #Edges | #Features | Avg. #Nodes | Avg. #Edges | Continuous Feature | Discrete Feature |
|---------|----------|---------|--------|--------|-----------|-------------|-------------|--------------------|--------------------|
| ENZYMES | 6 | 600 | - | - | 21 | 32.63 | 62.14 | ✓ | ✓ |
| DHFR | 2 | 756 | - | - | 56 | 42.43 | 44.54 | ✓ | ✓ |
| PROTEINS | 2 | 1113 | - | - | 4 | 39.06 | 72.82 | ✗ | ✓ |
| BACE | 2 | 1513 | - | - | 9 | 34.1 | 73.7 | ✗ | ✓ |
| BBBP | 2 | 2050 | - | - | 9 | 23.9 | 51.6 | ✗ | ✓ |
| Cornell | 5 | 1 | 183 | 298 | 1703 | - | - | ✗ | ✓ |
| Texas | 5 | 1 | 183 | 325 | 1703 | - | - | ✗ | ✓ |
| Wisconsin | 5 | 1 | 251 | 515 | 1703 | - | - | ✗ | ✓ |
| Cora | 7 | 1 | 2708 | 10556 | 1433 | - | - | ✗ | ✓ |
| CiteSeer | 6 | 1 | 3327 | 9104 | 3703 | - | - | ✗ | ✓ |
| PubMed | 3 | 1 | 19717 | 88648 | 500 | - | - | ✗ | ✓ |
| Flickr | 7 | 1 | 89250 | 899756 | 500 | - | - | ✗ | ✓ |

with some patterns. We study different distribution shifts in the graph domain to make a comprehensive evaluation for the setting of UGPP. Here is a brief review of these distribution shifts.

**Node-level.** Previous works study two main categories of distribution shift for nodes: 1- based on added random noise to node features (Knyazev et al., 2019) 2- based on structural properties such node degrees (Gui et al., 2022; Yuan et al., 2023), clustering coefficient (Bazhenov et al., 2023), Page Rank (PR) (Bazhenov et al., 2023), and Personalized Page Rank (PPR) (Zhu et al., 2021; Bazhenov et al., 2023). Because applying random noise to node features cannot necessarily represent distribution shifts, we use structural properties in our work. For that, we choose PR for main experiments as a popularity-based property since it implies challenging distribution shift (Zhu et al., 2021; Bazhenov et al., 2023). We also study the clustering coefficient as a density-based property and provide results.

**Graph-level.** Apart from datasets with inherently shifted distributions (Wu et al., 2017; Ying et al., 2019; Hu et al., 2020; Ding et al., 2021; Wu et al., 2022; Gui et al., 2022), number of nodes (Sui et al., 2023; Li et al., 2023), average node degrees (Li et al., 2022) and other graph properties can be utilized for introducing distribution shift for graph-level tasks. However, for main experiments, we choose edge homophily ratio (Zhu et al., 2020) for generating graph datasets with distribution shift because GNNs are intrinsically affected by this property for information aggregation (Gilmer et al., 2017; Zhu et al., 2020; Pei et al., 2020; Lim et al., 2021), and also our experiments on common graph classification datasets show higher variance of homophily ratio among graphs of these datasets. Additionally, we evaluate our method against graph density as another distribution shift as well.

### A.5.3 Experimental Details

**Task Unification.** A common practice for prompt tuning and in-context learning methods in graph domains is unification of different tasks (Sun et al., 2022; 2023; Huang et al., 2023). However, defining a unified task in the graph domain is more challenging than in text (e.g., using LLMs), for two main reasons. First, proposing a unified task requires an enormous amount of data, labeled or unlabeled, while, in contrast to text gathering, these large graph datasets are not feasible. Second, there is a wide variety of downstream graph tasks. Nonetheless, we unify the tasks to graph classification (Sun et al., 2023), considering the message-passing intrinsic of GNN (Gilmer et al., 2017).

To reduce node classification to graph classification, we select the induced subgraph of each ego node within its $k$-hop neighborhood and assign the label of the ego node to this subgraph. One can also reduce edge-level tasks by selecting the $k$-hop neighborhoods around the nodes lying on the endpoints of each edge (Sun et al., 2023).

**Implementation Details.** We have implemented our experiments in Pytorch (Paszke et al., 2017) and used a single GPU core NVIDIA GeForce RTX 3090. To make results reliable, we run each experiment

Table 9: Comparison against methods specialized for few-shot learning on graph classification under distribution shift. UGPROMPT (with 0% labels) consistently outperforms few-shot competitors (with 25% labels), which often suffer catastrophic negative transfer.

| Method | %Label | ENZYMES | PROTEINS | DHFR | BBBP | BACE |
|---|---|---|---|---|---|---|
| DAGPrompt | | -47.6 | -36.4 | -9.3 | -3.1 | -7.0 |
| PRONOG | 25 | -26.8 | -26.6 | -2.1 | 0.0 | 9.8 |
| GCOPE | | -28.5 | -15.8 | -6.2 | -3.5 | -15.2 |
| **UGPrompt** (ours) | **0** | **2.9** | **8.1** | **2.0** | **1.1** | **10.4** |

with 10 different random initializations of seeds before dataset creation and with 5 trials of model parameter initialization for every seed, which sums up to a total of 50 runs for every experiment.

We split all datasets based on properties to make a distribution shift to a 50% source dataset and a 50% target dataset. Since we split source and target datasets in the beginning in favor of making distribution shifts, we randomly make our own train, validation, and test splits for every trial, although the node classification datasets have original splits. Therefore, the train, validation, and test split is set to 0.6, 0.1, 0.3 for graph classification datasets and to 0.3, 0.1, 0.6 for node classification datasets—to reflect more on their original splits as they have many more test examples compared to train ones. Besides, for node classification, we find the induced graphs within nodes selected from datasets after the source-target split.

We tune hyper-parameters based on the average F1 score on validation sets as follows. For we select learning rate from $\{0.01, 0.001\}$, batch size from $\{16, 32, 64\}$, number of epochs from $\{30, 50, 60\}$, loss function weights for domain adaptation and diversity $(\lambda_1, \lambda_2)$ from $\{0.25, 0.5, 0.75, 1.0, 1.25, 1.5\}$, the $L_2$ regularization factor $(\lambda_3)$ from $\{0.1, 0.2\}$, the augmentation probability $p_u$ from $\{0.1, 0.2, 0.3, 0.4, 0.5\}$ and for $p_w$ from $\{0.05, 0.1, 0.2\}$, the number of trainable prompting parameter vectors $n_p$ from $\{10, 20, 30, 50, \mathbb{E}_{N_G}\}$ where $\mathbb{E}_{N_G}$ is the average number of nodes in graphs for the graph datasets. For the certainty threshold $\tau$, we either chose a fixed threshold following FixMatch (Sohn et al., 2020) or a dynamic class-wise threshold following FlexMatch (Zhang et al., 2021), then we select the threshold from $\{0.1, 0.3, 0.5, 0.7\}$. The final selection of all hyper-parameters for the GCN as base GNN and for main distribution shifts (edge homophily and PR) is provided in the codes provided by the link before.

### A.6 Additional Experimental Results

#### A.6.1 Comparison with More baselines

To validate whether UGPROMPT with no label constraint can offer practical advantages over few-shot methods, in this section, we evaluate our method against more recent state-of-the-art models, namely DAG-Prompt, PRONOG, and GCOPE. As evidenced in Tables 9 and 10, UGPROMPT generally outperforms the few-shot baselines in new experiments.

Specifically, UGPROMPT consistently surpasses all baselines on graph classification and achieves competitive (first or second best) results on node classification. The robustness of zero-label acts as a powerful regularizer, forcing distribution alignment for positive transfer, which contrasts sharply with few-shot methods that often suffer from catastrophic negative transfer and overfitting to biased samples under distribution shifts. In addition, UGPROMPT has notably less complex (lower number of trainable parameters) than the baselines. Therefore, heavier models like DAGPrompt (which has separate trainable low-rank matrices for each input and GNN encoder layer) fail catastrophically on smaller graph classification datasets—despite having superior performance on large and heterophilic node classification tasks—and undesirably show negative improvement as it requires a larger number of samples.

#### A.6.2 Effectiveness across Different Distribution Shifts.

The problem setup UGPP validates how well prompting methods can relieve the performance drop of a GNN facing distribution shift. Therefore, we aim to show the generalization of UGPROMPT across different kinds

Table 10: Comparison against methods specialized for few-shot learning on node classification under distribution shift. Few-shot methods achieve the highest gain on most datasets, but UGPROMPT is the best overall performer without relying on any labeled data.

| Method | %Label | Cora | CiteSeer | PubMed | Flickr | Cornell | Texas | Wisconsin |
|---|---|---|---|---|---|---|---|---|
| DAGPrompt | | 3.2 | 1.0 | **16.3** | **16.4** | **25.1** | **37.7** | **15.1** |
| PRONOG | 25 | -7.1 | -3.9 | -5.3 | -26.7 | -3.7 | 34.3 | 4.4 |
| GCOPE | | 0.4 | -4.5 | -2.4 | -30.9 | -14.1 | -6.8 | -3.9 |
| **UGPrompt (ours)** | **0** | **6.5** | **3.6** | 7.2 | 6.1 | 21.5 | 13.6 | 11.1 |

Table 11: Graph classification results for GCN as the base model on target datasets having distribution shift based on graph density. Overall, the best results are attained by UGPROMPT without any labels of the target dataset compared to the case where baselines have access to 50% of labels.

| Method | %Label | ENZYMES | | PROTEINS | | DHFR | |
|---|---|---|---|---|---|---|---|
| | | F1 | IMP | F1 | IMP | F1 | IMP |
| BaseModel | 0 | $39.1^{\pm 4.1}$ | 0.0 | $56.1^{\pm 4.0}$ | 0.0 | $76.3^{\pm 3.8}$ | 0.0 |
| Fine-Tuning | | $\mathbf{40.7}^{\pm 0.7}$ | **4.1** | $39.5^{\pm 1.6}$ | -29.6 | $76.6^{\pm 0.2}$ | 0.4 |
| GraphPrompt | 50 | $33.6^{\pm 1.4}$ | -14.1 | $55.7^{\pm 0.9}$ | -0.7 | $58.3^{\pm 3.1}$ | -23.6 |
| All-In-One | | $30.0^{\pm 3.2}$ | -23.3 | $37.3^{\pm 9.5}$ | -33.5 | $\underline{77.2}^{\pm 0.7}$ | 1.2 |
| GPF-Plus | | $39.2^{\pm 1.4}$ | 0.3 | $\underline{57.6}^{\pm 1.2}$ | 2.7 | $74.2^{\pm 0.9}$ | -2.8 |
| Fine-Tuning | | $38.0^{\pm 1.1}$ | -2.8 | $37.0^{\pm 3.6}$ | -34.0 | $73.6^{\pm 0.5}$ | -3.5 |
| GraphPrompt | 25 | $29.9^{\pm 1.4}$ | -23.5 | $53.3^{\pm 1.8}$ | -5.0 | $57.8^{\pm 3.1}$ | -24.2 |
| All-In-One | | $27.3^{\pm 3.8}$ | -30.2 | $34.2^{\pm 10.7}$ | -39.0 | $77.1^{\pm 0.7}$ | 1.0 |
| GPF-Plus | | $39.1^{\pm 1.3}$ | 0.0 | $57.2^{\pm 1.7}$ | 2.0 | $74.2^{\pm 0.9}$ | -2.8 |
| UGPROMPT (ours) | 0 | $\underline{40.0}^{\pm 1.0}$ | 2.3 | $\mathbf{58.3}^{\pm 0.8}$ | **3.9** | $\mathbf{78.0}^{\pm 0.8}$ | **2.2** |

of distribution shifts. Previously, we evaluated our method on distribution shift based on edge homophily ratio and Page Rank (PR). Here we add shifts based on graph density for graph classification and clustering coefficient for node classification.

*Graph Density Distribution Shift.* Table 11 illustrates UGPROMPT generally attains the best results on the graph datasets with consistent positive performance gain, while the competitors mostly have negative performance gain on DHFR and PROTEINS in all cases even when they take advantage of 50% of labeled data of the target distribution. Besides, we have the second-best performance on ENZYMES after Fine-tuning in 50% label setting while beating this baseline in 25% label setting.

*Node Clustering Coefficient Density Distribution Shift.* The results in Table 12 reflect on the superiority of UGPROMPT over all competitors on CiteSeer and PubMed. However, GPF-Plus outperforms our method on Cora when it has access to 50% of labeled samples while is beaten by UGPROMPT when 25% of labels are available. Notably, we excel over all of the baselines on PubMed with a high margin and with no labels.

### A.6.3 Generalization to Advanced GNN Backbones

A core principle of a prompting method is that it should be agnostic to the specific GNN backbone, demonstrating effectiveness across a range of architectures. To validate this, we extend our experiments beyond the foundational GCN and GAT models by incorporating two powerful, recent GNNs. We specifically chose: 1) GATv2 (Brody et al., 2022), a model that uses dynamic attention to better handle graphs with varying levels of homophily and heterophily. 2) GraphGPS (Rampasek et al., 2022), a graph transformer that uses a global attention mechanism to address common GNN challenges like oversmoothing and oversquashing.

The results of these new experiments are presented in Table 13 for graph and node classification tasks, respectively. The findings clearly demonstrate that our method, UGPROMPT, consistently outperforms all baselines when applied to these advanced GNN backbones. This is particularly noteworthy as the baselines retain the significant advantage of access to labeled data from the target domain, whereas our method operates in a completely unsupervised manner. These results confirm that the effectiveness of our prompting

Table 12: Node classification results for GCN as the base model on target datasets having distribution shift based on node clustering coefficient. When 50% of target dataset labels are visible to baseline, the unsupervised UGPROMPT achieves the second-best results while it performs the best overall datasets in cases where 25% of labels are available for competitors.

| Method | %Label | Cora | | CiteSeer | | PubMed | |
|---|---|---|---|---|---|---|---|
| | | F1 | IMP | F1 | IMP | F1 | IMP |
| BaseModel | 0 | $59.0^{\pm 3.1}$ | 0.0 | $44.1^{\pm 1.6}$ | 0.0 | $60.0^{\pm 0.8}$ | 0.0 |
| Fine-Tuning | | $60.2^{\pm 0.9}$ | 2.0 | $38.7^{\pm 0.5}$ | -12.2 | $53.5^{\pm 2.3}$ | -10.8 |
| GraphPrompt | | $59.9^{\pm 0.3}$ | 1.5 | $44.8^{\pm 0.3}$ | -1.6 | $61.3^{\pm 0.1}$ | 2.2 |
| GPPT | 50 | $47.9^{\pm 6.8}$ | -18.8 | $39.0^{\pm 2.2}$ | -11.6 | $55.1^{\pm 3.7}$ | -8.2 |
| All-In-One | | $53.7^{\pm 1.2}$ | -9.0 | $39.4^{\pm 1.0}$ | -10.7 | $47.1^{\pm 0.7}$ | -21.5 |
| GPF-Plus | | $\mathbf{61.4}^{\pm 0.6}$ | **4.1** | $41.7^{\pm 0.6}$ | -5.6 | $61.3^{\pm 1.0}$ | 2.2 |
| Fine-Tuning | | $56.5^{\pm 0.6}$ | -4.2 | $39.9^{\pm 0.4}$ | -9.5 | $48.7^{\pm 3.9}$ | -18.8 |
| GraphPrompt | | $58.0^{\pm 0.4}$ | -1.7 | $43.7^{\pm 0.3}$ | -0.9 | $61.3^{\pm 0.1}$ | -2.2 |
| GPPT | 25 | $46.4^{\pm 4.6}$ | -21.4 | $38.2^{\pm 2.9}$ | -13.4 | $54.5^{\pm 3.6}$ | -9.2 |
| All-In-One | | $53.7^{\pm 1.0}$ | -9.0 | $38.3^{\pm 0.9}$ | -13.2 | $48.9^{\pm 0.8}$ | -18.5 |
| GPF-Plus | | $59.8^{\pm 0.5}$ | 1.4 | $40.3^{\pm 0.6}$ | -8.6 | $60.9^{\pm 0.8}$ | 1.5 |
| UGPROMPT (ours) | **0** | $60.5^{\pm 0.3}$ | 2.5 | $\mathbf{45.1}^{\pm 0.4}$ | **2.3** | $\mathbf{64.7}^{\pm 0.3}$ | **7.8** |

Table 13: Graph classification results for GraphGPS and GATv2 as the base model on target datasets having distribution shift based on node clustering coefficient. UGPROMPT achieves almost the best across all datasets without labels, while 25% of labels are available for competitors.

| BaseGNN | Method | %Label | BBBP | | BACE | | Cornell | | Texas | | Wisconsin | |
|---|---|---|---|---|---|---|---|---|---|---|---|---|
| | | | F1 | IMP | F1 | IMP | F1 | IMP | F1 | IMP | F1 | IMP |
| | BaseModel | 0 | 87.0 | - | 73.0 | - | 22.8 | - | 24.4 | - | 20.8 | - |
| GraphGPS | Fine-Tuning | | 87.8 | 0.8 | 70.0 | -4.1 | 24.8 | 8.8 | 27.0 | 10.7 | 19.6 | -5.8 |
| | GraphPrompt+ | | 77.8 | -10.6 | 51.5 | -29.5 | 12.7 | -44.3 | 22.8 | -6.6 | 5.9 | -71.6 |
| | All-In-One | 25 | 72.1 | -17.1 | 39.5 | -46.9 | 19.6 | -14.0 | 0.3 | -98.8 | 22.3 | 7.2 |
| | GPF-Plus | | 87.7 | 0.8 | 73.6 | 0.8 | 22.9 | 0.4 | 26.9 | 10.2 | 22.5 | 8.2 |
| | UGPROMPT (ours) | **0** | **88.6** | **1.8** | **75.0** | **2.7** | **26.6** | **16.7** | **28.2** | **15.6** | **23.1** | **11.1** |
| | BaseModel | 0 | 88.8 | - | 63.8 | - | 16.0 | - | 22.0 | - | 24.7 | - |
| GATv2 | Fine-Tuning | | 89.4 | 0.7 | 67.8 | 6.3 | 17.8 | 11.3 | 22.3 | 1.4 | 23.5 | -4.9 |
| | GraphPrompt+ | | 83.6 | -5.9 | 64.3 | 0.8 | 10.3 | -35.6 | 2.0 | -90.9 | 8.2 | -66.8 |
| | All-In-One | 25 | 88.5 | -0.3 | 47.4 | -25.7 | 18.6 | 16.3 | 16.6 | -24.5 | 16.9 | -31.6 |
| | GPF-Plus | | 89.3 | 0.6 | **68.6** | **7.5** | 18.2 | 13.8 | 21.8 | -0.9 | 25.4 | 2.8 |
| | UGPROMPT (ours) | **0** | **89.6** | **0.9** | 67.9 | 6.4 | **20.2** | **26.3** | **23.0** | **4.5** | **26.2** | **6.1** |

framework is independent of a specific GNN architecture and that it generalizes robustly to more powerful and modern backbones.

### A.6.4 Comparison with Source-Free Domain Adaptation Methods

Since UGPP is a restrictive prompt-based setting within the broader SFDA landscape, the shared goal of adapting a model without source data motivates a direct empirical comparison. To this end, we evaluate UGPROMPT against two recent, state-of-the-art graph SFDA methods: SOGA(Mao et al., 2024b) and GraphCTA(Zhang et al., 2024). Since these methods are designed for node classification, we conduct the evaluation on our node classification datasets.

To create a fair comparison within our prompting-focused problem setup, we adapt these baselines. SFDA methods typically fine-tune the entire model; instead, we align them with the "lightweight fine-tuning" paradigm common to other baselines by freezing the source-trained GNN's encoder and only allowing the decoder (prediction head) to be trained on the target data. This contrasts with our method, UGPROMPT, where both the encoder and decoder remain fully frozen.

Table 14: Comparison again unsupervised SFDA methods. UGPROMPT outperforms both competitors while it has significantly less number of trainable parameters.

| Method | Label | Cora | | CiteSeer | | PubMed | | Flickr | |
|---|---|---|---|---|---|---|---|---|---|
| | | F1 | IMP | F1 | IMP | F1 | IMP | F1 | IMP |
| Base Model | - | 53.8 | - | 44.1 | - | 57.1 | - | 16.5 | - |
| SOGA | 0% | 53.1 | -1.3 | 43.7 | -0.9 | 54.9 | -3.9 | 14.5 | -12.1 |
| GraphCTA | | 49.2 | -8.6 | 38.0 | -13.8 | 12.1 | -78.8 | OOM | OOM |
| UGPROMPT (ours) | 0% | **57.3** | **6.5** | **45.7** | **3.6** | **61.2** | **7.2** | **17.5** | **6.1** |

Table 15: Comparison of IMP% against supervised prompting baselines with fine-tuned decoders using 25% labeled target data. GPPT is not applicable (NA) to graph classification. UGPROMPT generally outperforms the baselines without labels or decoder fine-tuning.

| Method | Texas | Cornell | Wisconsin | Cora | CiteSeer | PubMed | ENZYMES | PROTEINS | DHFR |
|---|---|---|---|---|---|---|---|---|---|
| GPPT | $9.2^{\pm32.8}$ | $-20.9^{\pm15.5}$ | $-8.1^{\pm14.7}$ | $-13.6^{\pm5.0}$ | $-12.6^{\pm3.9}$ | $-9.8^{\pm2.4}$ | NA | NA | NA |
| GraphPrompt | $23.1^{\pm39.3}$ | $-5.2^{\pm32.7}$ | $-11.9^{\pm21.0}$ | $-0.8^{\pm6.8}$ | $-3.5^{\pm4.0}$ | $-0.5^{\pm1.9}$ | $-21.2^{\pm9.2}$ | $-1.9^{\pm21.7}$ | $-6.1^{\pm9.5}$ |
| GraphPrompt+ | $27.5^{\pm35.2}$ | $12.0^{\pm34.6}$ | $-0.6^{\pm15.0}$ | $-9.0^{\pm8.1}$ | $-9.5^{\pm3.6}$ | $8.7^{\pm2.1}$ | $-50.0^{\pm7.0}$ | $-13.7^{\pm23.9}$ | $-14.7^{\pm6.8}$ |
| All-In-One | $-11.8^{\pm20.5}$ | $-34.5^{\pm14.3}$ | $-36.5^{\pm12.8}$ | $-10.6^{\pm18.4}$ | $-19.8^{\pm9.5}$ | $-10.0^{\pm11.9}$ | $-24.2^{\pm9.1}$ | $-25.6^{\pm26.7}$ | $-0.4^{\pm5.1}$ |
| GPF-Plus | $24.9^{\pm32.7}$ | $5.8^{\pm30.4}$ | $4.7^{\pm25.3}$ | $-1.7^{\pm5.3}$ | $-3.0^{\pm3.9}$ | $10.2^{\pm2.3}$ | $-17.3^{\pm12.0}$ | $-17.7^{\pm20.3}$ | $2.2^{\pm4.9}$ |
| UGPROMPT | $13.6^{\pm23.5}$ | $23.6^{\pm16.2}$ | $12.7^{\pm10.5}$ | $2.5^{\pm4.8}$ | $1.3^{\pm3.5}$ | $6.1^{\pm2.6}$ | $0.8^{\pm9.6}$ | $6.9^{\pm16.2}$ | $1.4^{\pm5.5}$ |

The results of this comparison are presented in Table 14. UGPROMPT consistently and significantly out-performs both adapted SFDA methods across all datasets. This outcome is powerful, as it demonstrates that our parameter-efficient approach of training only a prompt is more effective for adaptation than the more common fine-tuning strategy for the prediction head. In addition, our method is significantly lighter since we only train a small set of trainable prompting vectors, but these methods are originally proposed to fine-tune all parameters of the pretrained model.

## A.7 Comparison with Decoder Fine-tuned Prompting Baselines

In the main experiments, we restrict all prompting baselines to the frozen-GNN setting required by UGPP. This allows us to isolate the effect of the prompting function without conflating it with fine-tuning the source-trained encoder or prediction head. However, several supervised GNN prompting methods were originally designed to fine-tune a task-specific decoder. To provide a more comprehensive comparison, we further evaluate the supervised baselines in this relaxed setting, where their decoders are fine-tuned using 25% labeled target data.

Table 15 reports the performance improvement over the frozen BaseModel. Even with access to labeled target data and decoder fine-tuning, the supervised baselines often suffer from negative transfer under distribution shift. In contrast, UGPROMPT keeps both the encoder and decoder frozen, uses no labels, and still achieves stable positive improvements in most cases. These results further support that the gains of UGPROMPT come from effective unsupervised prompting rather than from additional fine-tuning.

### A.7.1 Embedding distribution analysis under distribution shift

Figure 3 illustrates the distribution of embeddings produced by the base GNN's encoder for the PubMed dataset. The setting is for PR distribution shift with GCN as the base GNN. It compares the embeddings of the BaseModels on source (BaseModel-source) and target (BaseModel-Target) test data, with graphs not prompted in either case. Additionally, it shows the embeddings for UGPROMPT and GPF-Plus when prompted with graphs from the target test data, while UGPROMPT is fully unsupervised and GPF-Plus consumes 50% of the labeled data.

This figure provides empirical evidence supporting our claims. First, UGPROMPT, which employs consistency regularization without labeled data, achieves equal or better performance in mitigating distribution shift compared to GPF-Plus, although the latter has access to labeled data. Second, when using confident pseudo-labels and keeping the GNN's parameters fixed after source-training, the prompting function aligns with the

knowledge learned from the source data. This approach preserves samples within densely homophilous regions while pushing uncertain samples (those in overlapping regions) away, replicating the source data distribution as seen in BaseModel-Source. As a result, the method produces well-separated representations, enabling the projection head to effectively discriminate between classes using the same source-trained weights.

### A.7.2 Effect of regularization methods

We study the effect of domain adaptation and diversity regularization methods introduced in Section 3.2. Minimizing the entropy of the score's expected value (as a diversity regularization term) has also been used in prior representation learning studies (Hu et al., 2017; Liang et al., 2020a). This regularization term relieves the harmful effect of class imbalance. Meanwhile, the domain adaptation regularization term is more beneficial for smaller datasets, since inferring their distribution from a small number of samples is difficult, increasing the likelihood of generating OOD prompted graphs. To showcase the effect of these objectives, we introduce two common methods of measuring class imbalance: the Imbalanced Ratio (IR) and Normalized Entropy (NE). A higher IR and a lower NE show more class imbalance in the datasets. Denoting the frequency of each class of a dataset as $f_i$, while we have $C$ classes, IR and NE are defined as below:

$$IR = \frac{f_{max}}{f_{min}}; \qquad f_{min} = \arg\min_i f_i, \quad f_{max} = \arg\max_i f_i \tag{7}$$

$$NE = \frac{H}{log(C)}; \qquad H = -\sum_{i=1}^{C} p_i log(p_i), \quad p_i = \frac{f_i}{\sum_{i=1}^{C} f_i} \tag{8}$$

Tables 16 shows the total number of samples in the target datasets (50% of the actual dataset sizes because we split the datasets into half source and half target), the statistics of their corresponding test sets, and the IMP% of UGPROMPT on these datasets (we fix GCN as the base GNN).

When looking at both graph and node classification results, the general trends support our claims that the regularization methods have a positive impact. The first main trend observed in each task group is that diversity regularization is more effective as dataset imbalance increases; specifically, on PROTEINS and Cora, which are the most imbalanced datasets for graph and node classification respectively, diversity regularization shows the highest IMP%. Furthermore, when considering dataset size, domain regularization generally demonstrates its greatest effect on smaller datasets. For example, Cora, the node classification dataset with the fewest samples, benefits the most from domain regularization. An exception regarding domain regularization's effect might be seen when comparing PROTEINS with DHFR and ENZYMES; this could be because PROTEINS has discrete features, whereas ENZYMES and DHFR also include continuous features, potentially reducing the impact of domain regularization.

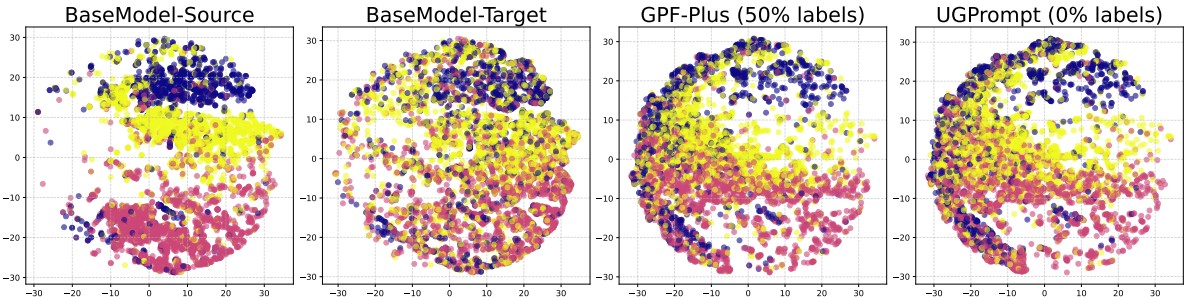

Figure 3: Distribution of embeddings generated by the base GNN encoder on the PubMed dataset under PR distribution shift with GCN as the base GNN. Embeddings for BaseModel on source and target test data, without prompting, and for UGPROMPT and GPF-Plus when target test graphs are prompted, are presented. This highlights UGPROMPT 's ability to mitigate distribution shifts without labeled data, producing well-separated representations that are similar to the source distribution.

Table 16: Statistic of the datasets used for experiments.

| Dataset | #Samples | NE | IR | IMP% of Diversity Reg. | IMP% of Domain Reg. |
|---|---|---|---|---|---|
| ENZYMES | 300 | 0.997 | 1.26 | 1.5 | 1.2 |
| PROTEINS | 556 | 0.949 | 1.72 | 6.6 | 4.3 |
| DHFR | 378 | 0.980 | 1.39 | 0.0 | 0.7 |
| Cora | 1354 | 0.941 | 4.72 | 4.0 | 0.7 |
| CiteSeer | 1663 | 0.965 | 1.94 | 1.3 | 0.0 |
| PubMed | 9858 | 0.966 | 1.91 | 0.2 | 0.0 |

Table 17: Ablation study on the contribution of each objective function ($\lambda_1$ and $\lambda_2$ are respectively the weights of diversity and domain adaptation objectives as in Equation 6) while using GCN as the base model. Both objectives show positive effect on UGPROMPT's performance.

| Dataset | Base ($\lambda_1$, $\lambda_2$=0.0) | | | | | | | |
|---|---|---|---|---|---|---|---|---|
| **Cora** | **F1=56.3** | $\lambda_1$ | 0.25 | 0.5 | 0.75 | 1.0 | 1.25 | 1.5 |
| | | **IMP%** | 1.4 | 1.6 | 2.0 | 2.0 | 2.0 | 2.0 |
| | | $\lambda_2$ | 0.25 | 0.5 | 0.75 | 1.0 | 1.25 | 1.5 |
| | | **IMP%** | 1.2 | 1.1 | 1.1 | 0.9 | 0.9 | 1.1 |
| **PROTEINS** | **F1=51.0** | $\lambda_1$ | 0.25 | 0.5 | 0.75 | 1.0 | 1.25 | 1.5 |
| | | **IMP%** | 3.1 | 5.9 | 7.1 | 7.5 | 7.5 | 7.5 |
| | | $\lambda_2$ | 0.25 | 0.5 | 0.75 | 1.0 | 1.25 | 1.5 |
| | | **IMP%** | 4.1 | 5.1 | 6.3 | 6.3 | 6.5 | 6.9 |

To better show the effects of each regularization method, we also conduct a more comprehensive ablation study on Cora and PROTEINS, which serve as representatives of datasets for node and graph classification. Table 17 shows how increasing the weight of each regularization term, as in Equation 6, improves the performance on both datasets, especially on PROTEINS.

### A.7.3 Effectiveness under no distribution shift

A key concern for any adaptation method is whether adapting to a new target distribution causes "catastrophic forgetting," degrading performance on the original source domain. To measure this, we adapted models on the target data (with a distribution shift) and then evaluated their performance on both the original source test set and the target test set. The results are presented in Table 18, with columns marked (S) for source and (T) for target evaluation. The findings show that UGPROMPT not only adapts effectively to the target distribution but also maintains higher performance on the source data compared to traditional lightweight fine-tuning, demonstrating its robustness against catastrophic forgetting.

To further understand our method's behavior, we also investigated its effect in a no-shift setting, where the model is prompted with a target domain that shares the same distribution as the source domain. As shown in Table 19, we measured the F1 score improvement (IMP) over the base model in this scenario. The results demonstrate that UGPROMPT consistently improves performance even without a distribution shift. This benefit is more evident than that of competing baselines, which show smaller gains and occasionally fail to improve performance at all. This indicates that our prompting approach serves as a general performance enhancer, not just a tool for mitigating distribution shifts.

### A.7.4 Effect of the number of trainable prompting vectors

As the final ablation study, we evaluate the effect of increasing the number of trainable prompting vectors. For this experiment, we also fix GCN as the base GNN and report the results in Table 20. This empirical evaluation clarifies that increasing the number of does not have a significant impact on UGPROMPT's performance. This conclusion is indeed favorable, meaning that our method can achieve desirable results even with a considerably low number of trainable parameters.

Table 18: Performance comparison of different methods across various source (S) and target (T) datasets.

| Method | %Label | Cora | | CiteSeer | | PubMed | | ENZYMES | | PROTEINS | | DHFR | |
|---|---|---|---|---|---|---|---|---|---|---|---|---|---|
| | | F1(S) | F1(T) | F1(S) | F1(T) | F1(S) | F1(T) | F1(S) | F1(T) | F1(S) | F1(T) | F1(S) | F1(T) |
| Base Model | - | 75.5 | 53.8 | 63.1 | 44.1 | **81.9** | 57.1 | 49.5 | 47.7 | 60.5 | 51.8 | 78.2 | 75.5 |
| Fine-Tuning | 25 | 69.8 | 51.7 | 57.2 | 40.0 | 74.2 | 54.3 | 43.4 | 46.4 | 56.2 | 47.5 | **79.4** | 76.8 |
| UGPROMPT (ours) | **0** | **75.7** | **57.3** | **63.3** | **45.7** | 81.0 | **61.2** | **49.6** | **49.1** | **63.3** | **56.0** | 78.5 | **77.0** |

Table 19: Performance comparison of UGPROMPT against the competitor under no distribution shift. An overall assessment across all datasets shows that our unsupervised method performs superiorly.

| Method | Label | Cora | | CiteSeer | | PubMed | | ENZYMES | | PROTEINS | | DHFR | |
|---|---|---|---|---|---|---|---|---|---|---|---|---|---|
| | | F1 | IMP | F1 | IMP | F1 | IMP | F1 | IMP | F1 | IMP | F1 | IMP |
| Base Model | | 64.9 | - | 50.5 | - | 73.2 | - | 54.5 | - | 51.8 | - | 75.1 | - |
| GraphPrompt+ GPF-Plus | 25 | **72.6** 66.5 | **11.9** 2.5 | 48.8 51.2 | -3.4 1.4 | **76.4** 73.6 | **4.4** 0.5 | 57.1 57.9 | 4.8 6.2 | 35.2 49.5 | -32.2 -4.4 | 55.4 76.1 | -26.3 1.3 |
| UGPROMPT (ours) | **0** | 65.9 | 1.5 | **51.4** | **1.8** | 75.8 | 3.6 | **58.7** | **7.7** | **52.2** | **0.8** | **76.2** | **1.5** |

Table 20: The effect of the number of trainable prompting vectors ($n_t$) while using GCN as the base model. A higher number of trainable vectors yields only a marginal improvement, and the model performs favorably with fewer trainable vectors.

| Dataset | $n_t = 10$ | $n_t = 20$ | $n_t = 30$ | $n_t = 40$ | $n_t = 50$ | $n_t = 60$ |
|---|---|---|---|---|---|---|
| **Cora** | 57.3 | 57.2 | 57.3 | 57.4 | 57.4 | 57.5 |
| **PROTEINS** | 55.5 | 56.0 | 56.1 | 55.8 | 56.7 | 56.2 |

### A.7.5 Analysis of Computational Cost

A practical consideration for any adaptation method is its computational cost. The unsupervised nature of UGPROMPT, which relies on data augmentation and multiple regularization components, introduces manageable overhead during training. This is an expected trade-off for the significant advantage of operating without labeled data. Table 21 shows the seconds of the average full dataset training epoch time and test time across node classification datasets. Our training times are marginally higher than supervised prompting baselines, but scale reasonably on large graphs like PubMed and Flickr.

However, the more critical metric for real-world deployment is inference efficiency. Once the prompt is trained, the adaptation process is complete. At test time, the expensive training components, such as data augmentation and the discriminator, are no longer required. The inference step simply involves a forward pass through the frozen GNN with the learned prompt, resulting in a computational complexity of $O(NLd^2 + L|E|d + Ndn_t)$—as discussed in Section 3.3—compared to $O(NLd^2 + L|E|d)$ of a regular GNN models and $n_t$ is number of trainable prompting vectors. Since, in all our experiments, even for large graphs $n_t \leq 50$, the overhead can be neglected. Our empirical results confirm this efficiency, showing that UGPROMPT's test time is consistently below the average of competing prompting methods, making it a lightweight and practical solution for deployment.

### A.7.6 Effect of changing the prompting function

An advantage of UGPROMPT, as discussed in Section 3.1, is its versatility, as it serves as a general framework agnostic to the base GNN and the prompting function. This means our unsupervised framework can potentially enhance a prefix prompting function such as (Fang et al., 2023; Sun et al., 2023). To support this claim, we design an experiment where we substitute our experimental prompting function (which is

Table 21: Comparison of UGPROMPT with other prompting methods based on the average full dataset training and test (inference) time measures in seconds. UGPROMPT has marginally higher training time and a test time below the average of all methods.

| Method | Cora | | CiteSeer | | PubMed | | Flickr | |
|---|---|---|---|---|---|---|---|---|
| | Test time (s), 813 nodes | Train time (s), 406 nodes | Test time (s), 984 nodes | Train time (s), 492 nodes | Test time (s), 5917 nodes | Train time (s), 2957 nodes | Test time (s), 22313 nodes | Train time (s), 17850 nodes |
| Fine-Tuning | 0.075 | 0.077 | 0.096 | 0.052 | 1.081 | 0.636 | 1.833 | 1.585 |
| All-In-One | 1.638 | 0.746 | 0.452 | 0.26 | 7.016 | 3.933 | 10.782 | 9.856 |
| GraphPrompt+ | 0.715 | 0.802 | 0.166 | 0.197 | 4.206 | 4.970 | 2.243 | 4.094 |
| GPF-Plus | 0.939 | 0.766 | 0.351 | 0.275 | 5.542 | 3.754 | 7.494 | 9.479 |
| UGPrompt (ours) | 0.925 | 1.181 | 0.315 | 0.458 | 5.56 | 4.946 | 6.730 | 13.904 |
| Average across Prompting Methods | 1.054 | 0.874 | 0.321 | 0.298 | 5.581 | 4.401 | 6.812 | 9.333 |

Table 22: Evaluation of UGPROMPT with All-In-One's prompting function using GCN as the base model. The results show that All-In-One's prompting function performs better when wrapped in our unsupervised framework, demonstrating that UGPROMPT is a versatile prompting framework that can enhance prefix prompting methods.

| Method | %Label | ENZYMES | | PROTEINS | | DHFR | |
|---|---|---|---|---|---|---|---|
| | | F1 | IMP | F1 | IMP | F1 | IMP |
| BaseModel | 0 | $47.7^{\pm 5.7}$ | - | $\mathbf{51.8}^{\pm 7.0}$ | - | $75.5^{\pm 6.3}$ | - |
| All-In-One | 50 | $\underline{48.7}^{\pm 1.0}$ | $\underline{2.1}$ | $45.8^{\pm 10.4}$ | -11.6 | $\underline{79.2}^{\pm 0.6}$ | $\underline{4.8}$ |
| | 25 | $45.8^{\pm 1.9}$ | -4.0 | $38.1^{\pm 13.4}$ | -26.4 | $79.1^{\pm 0.6}$ | 4.6 |
| UGPROMPT (ours) | **0** | $\mathbf{48.9}^{\pm 0.9}$ | **2.5** | $\underline{50.8}^{\pm 2.5}$ | **-1.9** | $\mathbf{79.3}^{\pm 2.4}$ | **5.0** |

similar to GPF-Plus) with All-In-One's prompting function and present the results in Table 22. These results show that All-In-One's prompting performs better when integrated into our framework. Notably, this improvement occurs without the use of labeled data.

**Remark on prompt tuning and other PEFT methods.** Other parameter-efficient fine-tuning (PEFT) methods for adaptation, such as LoRA, may also be considered for source-trained GNN adaptation. However, input-level prompt tuning is more aligned with the UGPP setting for two reasons. First, it treats the source-trained GNN as a frozen, potentially inaccessible model, since the prompt only modifies the input graph representation and does not require accessing or changing its internal layers. In contrast, methods such as LoRA inject trainable parameters into specific layers and therefore require architectural access to the model. Second, our consistency-based objective compares a weakly augmented graph with a strongly augmented and prompted graph, both of which are processed by the same frozen GNN. This keeps adaptation isolated in the prompting function. Moreover, standard GNNs are usually much smaller than LLMs, so layer-wise adapters may add a non-negligible number of parameters compared to our small set of trainable prompting vectors, which is important when adaptation relies only on unlabeled target data. We leave a systematic study of other PEFT methods under UGPP as future work.

### A.7.7 Effect of types of augmentation

Augmentation is a key component of our framework. As discussed in Section 3.1, the type of augmentation should align with the prompting function. For instance, if the prompting function applies feature modifications, as in our main experimental prompting function, feature augmentation is expected to be more beneficial than structural augmentation (e.g., adding or removing edges), and vice versa.

As an ablation study on the type of augmentation, Table 23 shows how different types of augmentation impact UGPROMPT's performance. Here, our prompting function transforms the feature matrix, the feature augmentation masks features, and structural augmentation drops edges. Results meet our expectations that

Table 23: Augmentation function's effect on UGPROMPT performance using GPF-Plus's feature prompting. Feature augmentation aligns more with feature prompting and generally achieves better results.

| Method | Aug. Type | ENZYMES | | PROTEINS | | DHFR | | Cora | | CiteSeer | | PubMed | |
|---|---|---|---|---|---|---|---|---|---|---|---|---|---|
| | | F1 | IMP | F1 | IMP | F1 | IMP | F1 | IMP | F1 | IMP | F1 | IMP |
| BaseModel | | 47.7 | - | 51.8 | - | 75.5 | - | 53.8 | - | 44.1 | - | 57.1 | - |
| UGPROMPT | Feature | **49.1** | **2.9** | **56.0** | **8.1** | **77.0** | **2.0** | **57.3** | **6.5** | **45.6** | **2.9** | **61.0** | **6.8** |
| (ours) | Structural | 48.2 | 1.0 | 54.5 | 5.2 | 75.2 | -0.4 | 57.0 | 5.9 | 44.8 | 1.6 | 55.7 | -2.5 |

Table 24: Source alignment of confidence-filtered target samples. We report MMD between source embeddings and confident/unconfident weakly augmented target samples. A lower MMD indicates closer alignment with the source distribution.

| Dataset | CiteSeer | Cornell | DHFR | Cora | Wisconsin | ENZYMES | PubMed | PROTEINS | Texas |
|---|---|---|---|---|---|---|---|---|---|
| Conf. Set | 0.026 | 0.118 | 0.117 | 0.055 | 0.126 | 0.072 | 0.024 | 0.394 | 0.099 |
| UNConf. Set | 0.119 | 0.257 | 0.381 | 0.172 | 0.299 | 0.075 | 0.726 | 0.262 | 0.214 |

feature augmentation improves performance, as it better aligns with the prompting function. Since none of the existing GNN prompting functions can be categorized solely as structural prompting (without changing the node representations), we leave experimenting with such a prompting method for future work.

## A.8 Source Alignment of Confidence-filtered Target Samples

Although the source data are not used during UGPROMPT training, we perform a diagnostic analysis to understand the role of confidence filtering. Therefore, source compatibility is induced indirectly through the frozen source-trained GNN and the confidence threshold used in $L_c$.

To verify this effect, we measure the Maximum Mean Discrepancy (MMD) between the latent embeddings of the source data and two subsets of weakly augmented target samples: confident samples with $\max(\tilde{\mathbf{p}}_\varphi^{\mathcal{G}}) \geq \tau$, and unconfident samples with $\max(\tilde{\mathbf{p}}_\varphi^{\mathcal{G}}) < \tau$. Table 24 shows that confident target samples are generally closer to the source distribution than unconfident ones. This supports our interpretation that confidence filtering selects the portion of the target distribution that is more compatible with the source-trained model.

Using the adversarial regularizer in $L_{\text{adv}}$ helps align the prompted graphs with their non-prompted weak augmentations on the target domain. According to the results, although this objective does not directly align the prompted representations with the unobserved source distribution, it prevents them from drifting too far from the target-side anchors, which are closer to the source samples.

## A.9 Pseudo-label Reliability and Sensitivity to Confidence Threshold

Pseudo-labeling under covariate shift may suffer from confirmation bias when high confidence predictions are incorrect. To evaluate this risk, we report pseudo-label accuracy, calibration before and after prompting, and sensitivity to the confidence threshold $\tau$ in Table 25. For binary classification datasets, we have $\tau \in \{0.6, 0.7, 0.8, 0.9\}$ and for multi-class classification we use $\tau \in \{0.2, 0.4, 0.6, 0.8\}$. First, increasing $\tau$ consistently improves pseudo-label accuracy, indicating that the threshold successfully filters out noisy predictions caused by distribution shift. Second, comparing the expected calibration error (ECE) before and after prompting reveals that UGPROMPT generally improves model calibration rather than amplifying overconfidence. Together, our regularized prompting framework extracts reliable signals and mitigates confirmation bias. We will include this analysis in the revised appendix.

### A.9.1 Unsupervised Prompting vs Fully Supervised Prompting

Since UGPROMPT achieves significant improvements across different experiments, we are interested in evaluating competitors while allowing them to access 50%, 75%, and 100% of the labeled data from the target

Table 25: Sensitivity of pseudo-label accuracy and calibration before and after prompting to the confidence threshold $\tau$. UGPROMPT shows that increasing the threshold generally improves pseudo-label precision.

| Dataset | Threshold $(\tau)$ | BaseModel Pseudo ACC ↑ | UGPrompt ECE ↓ | BaseModel ECE ↓ |
|---------|-----------|------------|---------|---------|
| Cornell | 0.2 | 0.289 | 0.141 | |
| | 0.4 | 0.302 | 0.144 | |
| | 0.6 | 0.314 | **0.135** | 0.138 |
| | 0.8 | **0.357** | 0.137 | |
| Wisconsin | 0.2 | 0.413 | 0.154 | |
| | 0.4 | 0.417 | 0.151 | |
| | 0.6 | 0.422 | **0.145** | 0.145 |
| | 0.8 | **0.439** | **0.145** | |
| Texas | 0.2 | 0.379 | 0.203 | |
| | 0.4 | 0.411 | 0.192 | |
| | 0.6 | **0.412** | 0.162 | 0.213 |
| | 0.8 | 0.359 | **0.161** | |
| ENZYMES | 0.2 | 0.538 | 0.189 | |
| | 0.4 | 0.560 | 0.188 | |
| | 0.6 | 0.579 | 0.189 | 0.185 |
| | 0.8 | **0.613** | **0.185** | |
| DHFR | 0.6 | 0.735 | **0.098** | |
| | 0.7 | 0.735 | **0.098** | |
| | 0.8 | 0.764 | 0.101 | 0.105 |
| | 0.9 | **0.825** | 0.101 | |
| PROTEINS | 0.6 | 0.698 | **0.082** | |
| | 0.7 | 0.698 | **0.082** | |
| | 0.8 | 0.724 | 0.083 | 0.092 |
| | 0.9 | **0.750** | 0.083 | |

distributions. We show the results of these settings in Tables 26 and 27, and Figures 4 and 5. All results are reported for both GCN and GAT under distribution shifts induced by edge homophily for graph classification and PR node classification.

Graph classification results for both GCN and GAT base models in Table 26 illustrate the superior performance of UGPROMPT as an unsupervised method compared to the baselines in most cases, even when they have access to 50% of the labeled samples for training on the target datasets. On the node classification datasets, UGPROMPT achieves the second-best performance as shown in Table 27. Specifically, for larger node classification datasets, it is noteworthy that we beat all baselines on Flickr and only underperform GraphPrompt+ on PubMed.

Observing Figure 4, when we use GAT as the base model, UGPROMPT outperforms GPF-Plus and Graph-Prompt on graph classification datasets even when they utilize fully labeled target datasets. *This clearly shows the effectiveness of our proposed method, UGPROMPT.* Additionally, we achieve closely competitive results with GCN as the base model in 100% label setting overall. Besides looking at the node classification results for both GNNs, we generally obtain the second-best improvement in 75% label setting and perform favorably with 100% compared to the best baseline.

Next, we evaluate our method in distribution shifts of graph density and clustering coefficient under the 75% and 100% label settings. Here, we fix GCN as the base GNN. Results are in Figure 5. Similar to the previous distributions, for graph density and clustering coefficient, UGPROMPT achieves competitive or better performance on graph classification datasets and ranks second after GPF-Plus on node classification, even though our method does not see any labels.

Table 26: Graph classification results on target datasets for GCN as the base model. Our unsupervised method mostly achieves the best results even when the competitors use 50% of the labeled data.

| Base GNN | Method | %Label | ENZYMES | | PROTEINS | | DHFR | |
|---|---|---|---|---|---|---|---|---|
| | | | F1 | IMP | F1 | IMP | F1 | IMP |
| GCN | BaseModel | 0 | $47.7^{\pm 5.7}$ | - | $51.8^{\pm 7.0}$ | - | $75.5^{\pm 6.3}$ | - |
| | Fine-Tuning | | $45.0^{\pm 2.0}$ | -5.7 | $46.9^{\pm 1.6}$ | -9.5 | $\underline{77.6}^{\pm 0.2}$ | $\underline{2.8}$ |
| | GraphPrompt | | $40.2^{\pm 1.5}$ | -15.7 | $\underline{54.1}^{\pm 1.0}$ | $\underline{4.4}$ | $73.1^{\pm 0.9}$ | -3.2 |
| | GraphPrompt+ | 50 | $29.7^{\pm 1.7}$ | -37.7 | $49.7^{\pm 1.0}$ | -4.1 | $65.5^{\pm 1.3}$ | -13.2 |
| | All-In-One | | $\underline{48.7}^{\pm 1.0}$ | $\underline{2.1}$ | $45.8^{\pm 10.4}$ | -11.6 | $\textbf{79.2}^{\pm 0.6}$ | $\textbf{4.8}$ |
| | GPF-Plus | | $48.6^{\pm 0.9}$ | 1.9 | $53.8^{\pm 2.4}$ | 3.9 | $77.4^{\pm 0.3}$ | 2.5 |
| | UGPROMPT (ours) | 0 | $\textbf{49.1}^{\pm 0.6}$ | 2.9 | $\textbf{56.0}^{\pm 1.5}$ | 8.1 | $77.0^{\pm 2.4}$ | 2.0 |
| GAT | BaseModel | 0 | $\underline{44.1}^{\pm 6.4}$ | - | $51.5^{\pm 8.1}$ | - | $77.3^{\pm 3.5}$ | - |
| | Fine-Tuning | | $43.3^{\pm 1.2}$ | -1.8 | $49.2^{\pm 0.5}$ | -4.5 | $\underline{78.0}^{\pm 0.2}$ | $\underline{0.9}$ |
| | GraphPrompt | | $35.3^{\pm 1.9}$ | -20.0 | $50.3^{\pm 0.9}$ | -2.3 | $77.1^{\pm 0.9}$ | -0.3 |
| | GraphPrompt+ | 50 | $29.2^{\pm 6.2}$ | -33.8 | $52.1^{\pm 6.3}$ | 1.2 | $60.5^{\pm 12.7}$ | -21.7 |
| | All-In-One | | $40.3^{\pm 2.5}$ | -8.6 | $41.8^{\pm 11.2}$ | -18.8 | $77.3^{\pm 1.2}$ | 0.0 |
| | GPF-Plus | | $43.4^{\pm 2.0}$ | -1.6 | $\underline{54.8}^{\pm 2.7}$ | $\underline{6.4}$ | $77.2^{\pm 0.8}$ | -0.6 |
| | UGPROMPT (ours) | 0 | $\textbf{45.9}^{\pm 2.2}$ | 4.1 | $\textbf{56.4}^{\pm 2.0}$ | 9.5 | $\textbf{78.2}^{\pm 0.9}$ | 1.2 |

Table 27: Node classification results on target datasets for GCN as the base model. Comparing all baselines with access to 50% of labeled data, UGPROMPT achieves the second-best results without labels.

| Base GNN | Method | %Label | Cora | | CiteSeer | | PubMed | | Flickr | |
|---|---|---|---|---|---|---|---|---|---|---|
| | | | F1 | IMP | F1 | IMP | F1 | IMP | F1 | IMP |
| GCN | BaseModel | 0 | $53.8^{\pm 2.4}$ | - | $44.1^{\pm 1.5}$ | - | $57.1^{\pm 0.8}$ | - | $\underline{16.5}^{\pm 0.4}$ | - |
| | Fine-Tuning | | $54.5^{\pm 1.2}$ | 1.3 | $43.5^{\pm 0.4}$ | -1.4 | $56.3^{\pm 2.1}$ | -1.4 | $10.3^{\pm 0.4}$ | -37.6 |
| | GPPT | | $50.5^{\pm 3.1}$ | -6.1 | $40.6^{\pm 1.2}$ | -7.9 | $51.8^{\pm 3.7}$ | -9.3 | $13.6^{\pm 0.5}$ | -17.6 |
| | GraphPrompt | | $55.8^{\pm 0.3}$ | 3.7 | $43.8^{\pm 0.3}$ | -0.7 | $57.1^{\pm 0.1}$ | 0.0 | $13.1^{\pm 0.2}$ | -20.6 |
| | GraphPrompt+ | 50 | $\underline{57.3}^{\pm 0.6}$ | $\underline{6.5}$ | $42.9^{\pm 0.2}$ | -2.7 | $\textbf{64.9}^{\pm 0.2}$ | $\textbf{13.7}$ | $14.9^{\pm 0.7}$ | -9.7 |
| | All-In-One | | $50.3^{\pm 1.2}$ | -6.5 | $39.3^{\pm 1.0}$ | -10.9 | $39.8^{\pm 1.2}$ | -30.3 | $13.5^{\pm 0.3}$ | -18.2 |
| | GPF-Plus | | $\textbf{58.2}^{\pm 0.6}$ | 8.2 | $\textbf{46.8}^{\pm 0.7}$ | 6.1 | $60.3^{\pm 0.5}$ | 5.6 | $13.1^{\pm 0.1}$ | -20.6 |
| | UGPROMPT (ours) | 0 | $\underline{57.3}^{\pm 0.4}$ | $\underline{6.5}$ | $\underline{45.7}^{\pm 0.4}$ | $\underline{3.6}$ | $\underline{61.2}^{\pm 0.3}$ | $\underline{7.2}$ | $\textbf{17.5}^{\pm 0.1}$ | 6.1 |
| GCN | BaseModel | 0 | $47.7^{\pm 1.3}$ | - | $41.2^{\pm 2.4}$ | - | $60.0^{\pm 1.1}$ | - | $\underline{17.0}^{\pm 0.2}$ | - |
| | Fine-Tuning | | $47.1^{\pm 1.7}$ | -1.3 | $39.9^{\pm 0.5}$ | -3.2 | $56.5^{\pm 2.2}$ | -5.8 | $10.8^{\pm 0.3}$ | -36.5 |
| | GPPT | | $32.8^{\pm 3.8}$ | -31.2 | $35.6^{\pm 1.2}$ | -13.6 | $51.8^{\pm 5.5}$ | -13.7 | $13.1^{\pm 0.3}$ | -22.9 |
| | GraphPrompt | | $48.2^{\pm 0.5}$ | 1.0 | $40.7^{\pm 0.3}$ | -1.2 | $60.1^{\pm 0.1}$ | 0.2 | $13.5^{\pm 0.1}$ | -20.6 |
| | GraphPrompt+ | 50 | $48.2^{\pm 0.5}$ | 1.0 | $42.0^{\pm 0.3}$ | 1.9 | $\textbf{67.1}^{\pm 0.1}$ | $\textbf{11.8}$ | $\textbf{17.5}^{\pm 0.6}$ | 2.9 |
| | All-In-One | | $34.6^{\pm 4.1}$ | -27.5 | $33.0^{\pm 1.2}$ | -19.9 | $25.4^{\pm 5.2}$ | -57.7 | $12.5^{\pm 0.5}$ | -26.5 |
| | GPF-Plus | | $\textbf{49.6}^{\pm 1.4}$ | 4.0 | $\textbf{43.1}^{\pm 0.6}$ | 4.6 | $60.1^{\pm 0.5}$ | 0.2 | $13.2^{\pm 0.2}$ | -22.4 |
| | UGPROMPT (ours) | 0 | $\underline{48.8}^{\pm 0.9}$ | $\underline{2.3}$ | $\underline{42.3}^{\pm 0.5}$ | $\underline{2.7}$ | $\underline{60.2}^{\pm 0.1}$ | $\underline{0.3}$ | $\textbf{17.5}^{\pm 0.1}$ | 2.9 |

Finally, an interesting finding of these experiments is that competitors can occasionally cause a performance drop compared to the base models, which is unexpected and undesirable. On the other hand, UGPROMPT, although it is an unsupervised method, does not negatively affect any dataset, any GNN architecture, or any type of distribution shift. Also, all the above results lead to the same conclusion: UGPROMPT can enhance base GNNs that encounter different distribution shifts across various downstream tasks. Since UGPROMPT achieves promising results in a fully unsupervised manner, it offers new avenues for leveraging large unlabeled datasets to improve the generalization of GNNs.

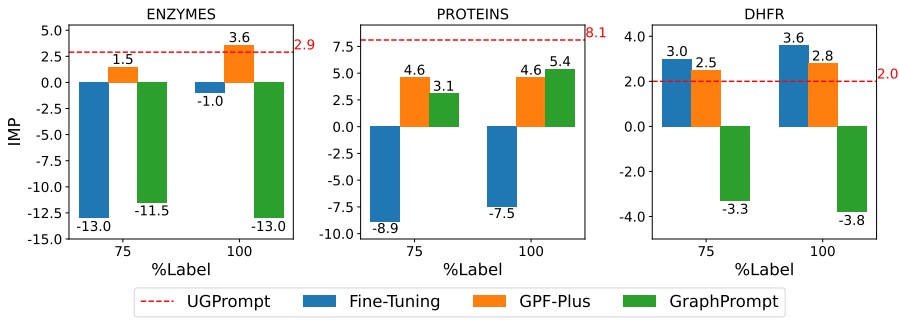

(a) Graph classification under edge homophily distribution shift for GCN as base model.

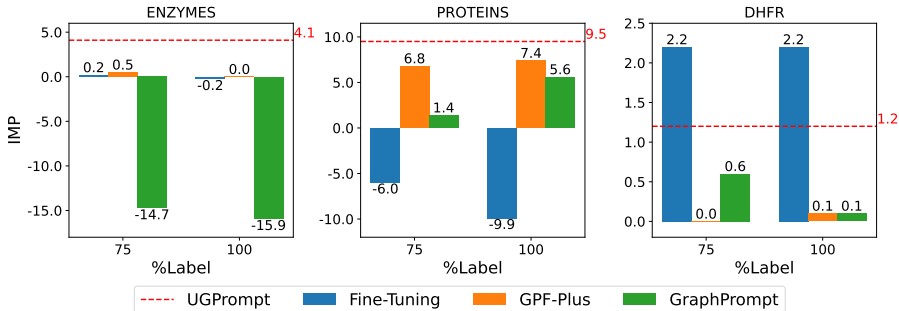

(b) Graph classification under edge homophily distribution shift for GAT as base model.

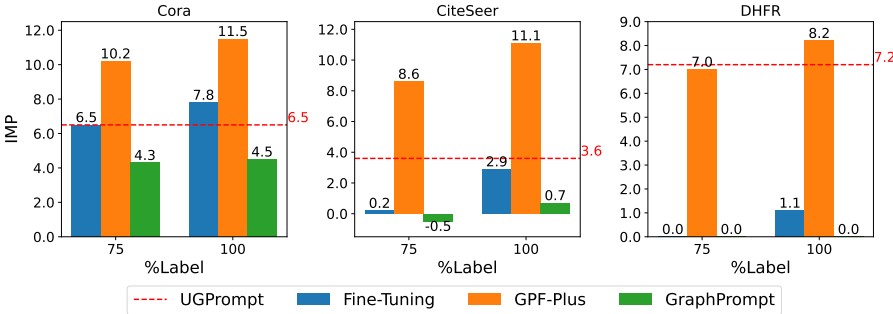

(c) Node classification under PR distribution shift for GCN as base model.

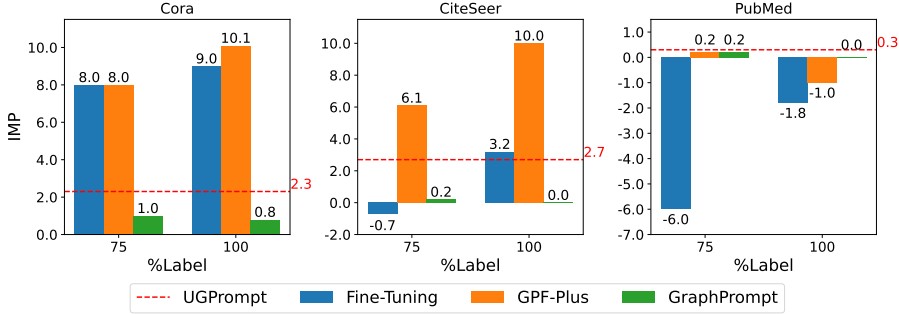

(d) Node classification under PR distribution shift for GAT as base model.

Figure 4: Performance gains for GCN and GAT base models on graph and node classification tasks in the presence of edge homophily (a, b) and node page rank (c, d) distribution shifts; where the competitor prompting methods utilize 100% and 75% labeled data of target distributions. UGPROMPT without using any labels always improves over the base model and achieves the best results on graph classification with GAT, while ranking second-best in other cases.

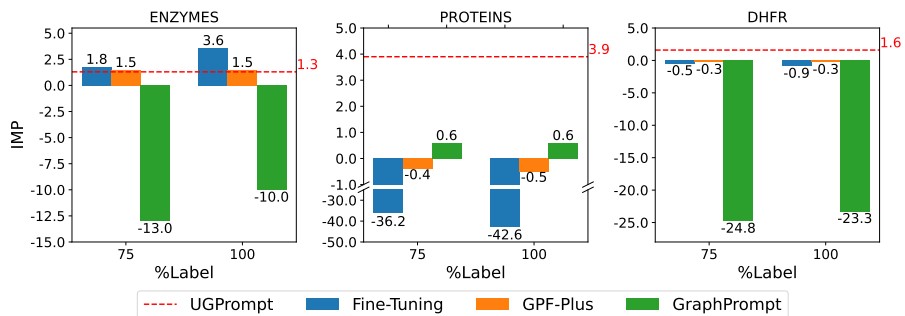

(a) Graph classification under graph density distribution shift for GCN as base model.

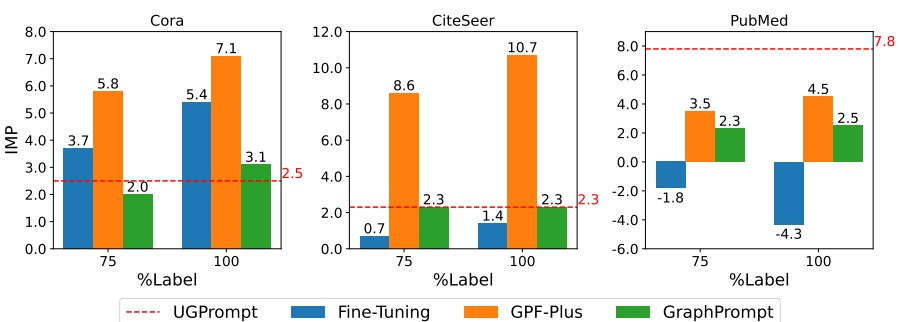

(b) Node classification under clustering coefficient distribution shift for GCN as base model.

Figure 5: Performance gains for GCN as the base model on graph and node classification tasks in the presence of graph density (a) and node clustering coefficient (b) distribution shifts when the competitor prompting methods utilize 100% and 75% labeled data of target distributions. The trend is similar to other distribution shifts (Figure 4) where UGPROMPT generally attains the best results on graph classification and the second-best on node classification.

