# OpenReview forum: "Freeze, Prompt, and Adapt: A Framework for Source-free Unsupervised GNN Prompting"
_TMLR — Accepted by TMLR_

### Review · Reviewer_eurC · 2026-03-06

**Summary Of Contributions:**

This paper introduces the Unsupervised Graph Prompting Problem (UGPP), a setting where a pre-trained GNN (including its projection head) is kept fully frozen, the source data is inaccessible, and adaptation must be performed on unlabeled target graphs under covariate shift. The proposed method, UGPrompt, trains only a prompting function by treating the prompted sample as a strong augmentation and using confident predictions on a weak augmentation as pseudo-labels, optimized via a consistency loss. To reduce collapse and mitigate distribution mismatch, it adds an entropy-based diversity regularizer and an adversarial discriminator that encourages prompted representations to resemble those of non-prompted augmentations. Experiments on several graph- and node-classification benchmarks report that UGPrompt often matches or exceeds multiple supervised prompting baselines trained with partial labels, under constructed shifts based on homophily or PageRank.

**Audience:**

Yes

**Audience Explanation:**

NA

**Claims And Evidence:**

Yes

**Claims Explanation:**

NA

**Requested Changes:**

1.	The problem definition and the experimental claims are internally inconsistent and sometimes ill-posed. UGPP explicitly assumes covariate shift with unchanged conditional label distribution (Pt(Y|X)=Ps(Y|X)) while freezing both encoder and projection head. Yet the paper highlights “label distribution shift with disjoint labels” and cross-dataset class splits where the base model is trained on classes {0,1,2} and evaluated on {2,3,4} (reindexed), which fundamentally changes the meaning of the classifier outputs and appears incompatible with a frozen head unless additional (unstated) label/weight remapping is performed. Compounding this, the paper later states UGPrompt is unable to handle label distribution shift because the projection head is not trained. These pieces cannot all be true as written.
2.	The novelty is thin relative to existing semi-supervised/unsupervised adaptation recipes, and the “prompting” component is largely a repackaging of feature-space adaptation. The core training signal is FixMatch-style pseudo-labeling plus consistency regularization, except applied to prompt parameters rather than model weights, and the prompt parameterization is simple additive token vectors similar to prior GNN prompting designs. The appendix explicitly frames the novelty as interpreting a prompted graph as a strong augmentation and using weak-augmentation pseudo-labels to train prompt parameters.
3.	Baselines are allowed access to labeled target data (e.g., 25%), while UGPrompt uses none, but the paper also restricts baselines by disallowing any updates to the GNN parameters and focusing only on their prompting modules. This alters the baselines from their intended training regimes (many are designed to train heads and/or lightly tune modules), so the reported “supervised SOTA” performance may not be representative. The main tables only emphasize 25% labeling for baselines, while higher-label settings are deferred to the appendix, making it hard to judge whether UGPrompt is truly competitive against properly tuned supervised approaches across label budgets.
4.	The pseudo-labeling mechanism is fragile under the very setting the paper claims to target, and the paper does not provide the necessary reliability diagnostics. The method selects pseudo-labels based on a confidence threshold applied to weak-augmentation predictions, then trains the prompt to match those labels on the prompted/strong version. Under covariate shift, high-confidence errors are common and can cause confirmation bias; it is therefore essential to report pseudo-label accuracy estimates, calibration before/after prompting, and sensitivity to tau, augmentation rates, and the class-wise/dynamic thresholding choice.
5.	The discriminator distinguishes prompted graphs from non-prompted weak augmentations, both drawn from the target distribution, and the prompt is trained to fool this discriminator. This encourages the prompted representation to match the weakly-augmented target representation, not to align with the (unobserved) source manifold. As a result, the claimed mitigation of OOD relative to source is speculative.

Minor concerns
1.	The paper claims efficiency advantages and reports a complexity expression, but provides no main-paper wall-clock/compute measurements or memory overhead breakdown including the discriminator and prompt parameters; runtime claims should be supported with actual numbers in the core results.
2.	Presentation issues: terminology is sometimes overloaded (“pre-train” vs supervised training on the source split for the same task), and the narrative repeatedly contrasts with SFDA while the setting is effectively a frozen-model source-free adaptation problem with an input transformation; tightening definitions would improve clarity.

---

> ### Author Response · Authors · 2026-04-03
> **Response to the Reviewer - Part I**
>
> >**W1: Clarification on consistency between shift assumptions, frozen head, and disjoint labels.**
>
> **Response**
>
> We sincerely thank the reviewer for the detailed comment. We understand the reviewer's point about the inconsistency between the label-shift experiment and the strict definition of UGPP ($P_t(Y|X)= P_S (Y|X)$), which requires a frozen projection head. However, as the projection head $\theta_h$ was strictly frozen after training on source classes {$0, 1, 2$}, its output dimension remained $C=3$. For the target dataset consisting of classes {$2, 3, 4$}, we manually mapped the ground-truth target labels to {$0, 1, 2$} solely for evaluation metrics. The model weights were never remapped or updated.
>
> We included this experiment as a more extreme test to see how our unsupervised prompting generalizes when the semantic meaning of the frozen classifier was entirely broken (resulting in the severe performance drop shown for the base model in Table 6).
>
> We acknowledge that framing this as "label distribution shift" is misleading. As stated in our Limitations, UGPrompt cannot natively solve true label shift because the projection head is not trained. In the revision, **we will rename this experiment to "Robustness under extreme category disjoint shift"** and clearly distinguish it from the core claims about UGPP to clarify this confusion.
>
> >**W2: Concerns about novelty**
>
> **Reponse**
>
> We thank the reviewer for their feedback. While the proposed method is inspired by consistency regularization techniques such as FixMatch, we emphasize that adapting these principles to a strictly frozen GNN via prompting is both non-trivial and novel.
>
> Specifically, our core contributions extend beyond repackaging as follows:
> *   **Novel Problem Formulation (UGPP):** Existing GNN prompting methods rely heavily on lightweight fine-tuning (updating the projection head) and require labeled target data. UGPP isolates the true efficacy of prompting by enforcing a strictly frozen pre-trained model and a source-free, unsupervised target domain. This strictly parameter-efficient paradigm is novel in the field of graph learning.
> *   **Methodological Novelty:** Applying consistency regularization to a frozen model requires careful structural design. We uniquely treat the *learnable prompt itself* as a strong, directed augmentation mechanism. Unlike standard FixMatch, which updates internal weights to learn invariance, our prompt explicitly transforms target input features to align with the frozen source learned manifold.
> *   **Targeted Regularization:** Because the GNN is frozen, standard pseudo-labeling often causes generating trivial and out-of-distribution graphs. To address these, we introduce the synergistic combination of our domain-adaptation discriminator and a diversity loss to enforce valid structural generation and prevent class collapse.
> *   **Extensive Empirical Validation:** As demonstrated across 12 datasets, 6 types of distribution shift (covariate and semantics), and against 11 SOTA baselines on 2 types of learning settings (unsupervised and few-shot), UGPrompt consistently achieves superior zero-shot adaptation, validating that this framework is a robust and novel solution to UGPP.
>
> >**W3: Evaluation fairness regarding baseline tuning regimes and labeling budgets**
>
> **Response**
>
> We appreciate this feedback. Our main comparison follows the UGPP setting, where the entire pre-trained GNN and decoder are frozen to isolate the prompt's impact under source-free, label-free adaptation. Allowing baselines to update the decoder would move beyond the UGPP scope and conflate prompt effects with head fine-tuning.
>
> While results for higher label budgets (50%–100%) in the appendix show similar trends of our method's superior performance, we have now added an experiment where supervised baselines can train the decoder at the 25% label ratio, like their original unrestrictive training regime. The results in Table R1 show that UGPrompt remains competitive and outperforms these baselines on most datasets. We will move this table into the main paper to show that our results do not depend on overly restricting the baselines. (GPPT is not applicable to graph classification, so we report NA for that)
>
> **Table R1: Comparison of Improvement of performance (IMP%) against baselines with fine-tuned decoders (25% labels).**
> |Method|CIT|CNL|COR|DHFR|ENZ|PRO|PUB|TEX|WIS|
> |-|-|-|-|-|-|-|-|-|-|
> |All-in-One|-19.8|-34.6|-10.6|-0.4|-24.2|-25.5|-10.0|-11.9|-36.5|
> |GPF-Plus|-2.9|5.8|-1.6|2.1|-17.4|-17.8|10.3|25.0|4.8|
> |GPPT|-12.5|-20.9|-13.6|NA|NA|NA|-9.8|9.3|-7.9|
> |GraphPrompt|-3.6|-5.2|-0.7|-6.0|-21.3|-1.9|-0.5|23.3|-11.9|
> |GraphPromptPlus|-9.6|12.0|-9.0|-14.7|-49.3|-13.7|8.7|27.5|-0.8|
> |UGPrompt (ours)|1.3|23.6|2.5|1.4|0.8|6.9|6.1|13.6|12.7|

---

> ### Author Response · Authors · 2026-04-03
> **Response to the Reviewer - Part II**
>
> >**W4: Pseudo-labeling reliability and lack of complete sensitivity analysis**
>
> **Response**
>
> We agree that pseudo-labeling under covariate shift can be prone to confirmation bias if high-confidence predictions are inaccurate. To address this, we have added the requested reliability diagnostics.
>
> **Table 3 of the paper** already includes an ablation on augmentation strength ($p_s$), showing how performance varies with the strength of augmentation. Regarding the threshold $\tau$, it is a key hyperparameter (either fixed or class-dynamic) that we set during tuning following standard SSL practices.
>
> To further support our findings, we have now added the three suggested analyses in **Table R2** below: pseudo-label accuracy estimates, calibration metrics before and after prompting, and sensitivity tests for the confidence threshold.
>
> The results demonstrate two key findings. **First,** as the threshold $\tau$ increases, pseudo-label accuracy consistently improves, confirming that our threshold successfully filters out noisy predictions caused by distribution shift. **Second,** comparing ECE before and after prompting reveals that UGPrompt generally improves model calibration rather than amplifying overconfidence. Together, it is shown that our regularized prompting framework extracts reliable signals and mitigates confirmation bias. We will include this analysis in the revised appendix.
>
> **Table R2: Pseudo-Label reliability and calibration across confidence thresholds**
> |Dataset|Conf. Threshold ($\tau$)|After Prompt ECE|Before Prompt ECE|Pseudo-label Accuracy|
> |-|-|-|-|-|
> |CON|0.2|14.1|13.8|28.9|
> |CON|0.4|14.4|13.8|30.2|
> |CON|0.6|13.5|13.8|31.4|
> |CON|0.8|13.7|13.8|35.7|
> |DHFR|0.2|9.8|10.5|73.5|
> |DHFR|0.4|9.8|10.5|73.5|
> |DHFR|0.6|10.1|10.5|76.4|
> |DHFR|0.8|10.1|10.5|82.5|
> |ENZ|0.6|18.5|18.9|53.8|
> |ENZ|0.7|18.5|18.9|56.0|
> |ENZ|0.8|18.5|18.9|57.9|
> |ENZ|0.9|18.5|18.5|61.3|
> |PRO|0.2|8.2|9.2|69.8|
> |PRO|0.4|8.2|9.2|69.8|
> |PRO|0.6|8.3|9.2|72.4|
> |PRO|0.8|8.3|9.2|75.0|
> |TEX|0.2|20.3|21.3|37.9|
> |TEX|0.4|19.2|21.3|41.1|
> |TEX|0.6|16.2|21.3|41.2|
> |TEX|0.8|16.1|21.3|40.9|
>
> >**W5: Speculative OOD mitigation and alignment of prompted representations.**
>
> **Response**
>
> We thank the reviewer for this insightful observation. It is true that the discriminator and consistency loss align the prompted graph ($\mathcal{G}_p$) with the weakly augmented target graph ($\mathcal{G}_w$). However, mitigating the OOD shift relative to the source is not speculative; it is achieved through our confidence threshold ($\tau$).
>
> Since the frozen base GNN was trained exclusively on source data, it acts as an anchor, naturally yielding high-confidence predictions only for target samples that are structurally and feature-wise similar to the source distribution. By applying the consistency loss ($\mathcal{L}_c$) only to samples exceeding this threshold, the prompt is forced to align strongly augmented graphs specifically with the subset of target data that already overlaps with the source manifold.
>
> To prove this empirically, in **Table R3**, we measured the Maximum Mean Discrepancy (MMD) between the latent embeddings of the unobserved source data and two subsets of the weakly augmented target data: the "Confident" samples (max score $\ge \tau$) and the "Unconfident" samples (max score $< \tau$).
>
> Across nearly all datasets, confident target samples exhibit a significantly smaller distance to the source distribution than unconfident samples (with PRO being outliers due to its specific discrete features). This confirms that high-confidence target samples lie near the source distribution, so by matching them, the prompt pulls OOD target graphs toward the source manifold. We will include this MMD analysis in the appendix.
>
> **Table R3: Source alignment of confidence-filtered target samples**
> |Dataset|Conf. Set(MMD)|UNConf. Set(MMD)|
> |-|-|-|
> |CIT|0.026|0.119|
> |CON|0.118|0.257|
> |DHFR|0.117|0.381|
> |COR|0.055|0.172|
> |WIS|0.126|0.299|
> |ENZ|0.072|0.075|
> |PUB|0.024|0.726|
> |PRO|0.394|0.262|
> |TEX|0.099|0.214|

---

> ### Author Response · Authors · 2026-04-04
> **Response to the Reviewer - Part III**
>
> >**Minor concerns**
>
> >**1. Runtime analysis**
>
> **Response**
>
> We thank the reviewer for highlighting the importance of explicitly supporting our efficiency claims with empirical measurements. The paper already includes a detailed runtime evaluation and theoretical complexity analysis in **the Appendix A.6.9 and Table 20**. We will add the following summaries to the main text to resolve the reviewer's concerns:
>
> * **Wall-clock runtime:** UGPrompt involves slightly longer training due to unsupervised consistency learning. However, it is highly efficient at inference; once training is complete, the augmentation pipelines and discriminator are removed, allowing the model to run as fast as the base GNN.
> * **Memory overhead:** Our prompting function adds extremely small number of trainable parameters (fewer than 50 low-dimensional vectors across all experiments). Also, the trainable discriminator is a lightweight 2-layer feed-forward network that is removed at inference time. This minimal overhead ensures practicality and scalability.
>
> >**2: Presentation issues**
>
> **Response**
>
> We appreciate the reviewer pointing out these details. To improve the clarity of the manuscript, we will make the following adjustments:
>
> * **Standardizing terminology:** We agree that using the term "pre-train" to describe supervised training on source data can be confusing. In the revised manuscript, we will uniformly use "supervised training on the source domain" to eliminate any ambiguity.
> * **Reframing relative to SFDA:** We agree with the reviewer's assessment regarding Unsupervised Source-Free Domain Adaptation (USFDA). Our original intent was to highlight the restrictive nature of our setting (a frozen GNN + input transformation), but we acknowledge that directly contrasting it with SFDA might be misleading. In the revision, we will tighten our definitions to explicitly position UGPP as a specific, restricted, and prompt-based variant within the broader USFDA framework, rather than a wholly distinct framework.

---

### Review · Reviewer_q58f · 2026-03-15

**Summary Of Contributions:**

This paper proposes a new task for unsupervised source-free domain adaptation for GNN models using a prompt-tuning technique inspired by LLMs. Instead of finetuning or updating the graph neural network itself, the authors introduce a prompting function that effectively "prompts the graph", enabling the GNN to better interpret data from a new target dataset.  Experiments conducted with GCN and GAT architectures show improvements on both node and graph classification tasks compared to supervised-finetuning methods and baseline models, suggesting the effectiveness of the approach as well as its model-agnostic nature.

**Additional Comments:**

* Some notations are not consistent. For example, the paper uses both "GNN" and "Gnn", and the abbreviation of the proposed problem "UGPP" sometimes appears as "Ugpp". It would be better to standardize these notations throughout the paper.
* The citation format appears somewhat inconsistent. The authors may want to consider using `\citep` instead of `\cite` in the source tex files to ensure the citation format is displayed correctly.

**Audience:**

Yes

**Audience Explanation:**

Graph neural networks are an important research topic in AI4Science, recommendation system, as well as in applications that analyze relationships among entities. These themes align well with the interests of the TMLR community.

**Broader Impact Concerns:**

The paper does not raise any broader impact concerns.

**Claims And Evidence:**

Yes

**Claims Explanation:**

* The paper clearly describes and compares the task setup in Section 2.2, with additional details provided in Appendix A.4, which helps readers understand the differences introduced by the new task formulation.
* The paper conducts experiments on two graph architectures and across different graph tasks (Table 1 and Table 2), demonstrating that the approach is (1) model-agnostic and (2) task-agnostic. In particular, the results outperform previous supervised fine-tuning methods.

**Requested Changes:**

I am not very familiar with the latest SOTA research in GNNs. From my perspective, the overall experimental results and comparisons appear clear and reasonably strong, and I do not have major concerns about the experimental design itself. However, the results are not entirely convincing to me.

* It is unclear why the unsupervised setup achieves better results than approaches that allow supervised training. From an information perspective, supervised training should provide more informative signals (e.g., more accurate gradient directions) and would typically be expected to serve as an upper bound compared to unsupervised methods. However, the reported results show noticeable improvements without additional discussion on this point. It would be helpful if the authors could provide an explanation for why the proposed approach is able to outperform methods that have access to supervised signals.
* Another question concerns the choice of prompt tuning. What are the advantages of using prompt tuning in this context? Could other parameter-efficient fine-tuning (PEFT) techniques, such as LoRA, also serve as effective universal adaptation methods? Similarly, LoRA weights can be merged into the original model during inference, resulting in no additional runtime cost. While the paper seems to follow prior work in adopting prompt tuning, it would still be valuable for the authors to discuss whether other PEFT approaches could also be suitable for this task and how prompt tuning compares to them.

---

> ### Author Response · Authors · 2026-04-03
> **Response to the Reviewer**
>
> >**W1: Lack of explanation for unsupervised performance exceeding supervised benchmarks.**
>
> **Response**
>
> We thank the reviewer for raising this point. In standard settings where the entire model is fine-tuned, supervised learning is an upper bound. However, under the specific setting of our problem, UGPP (frozen GNN + distribution shift), this no longer holds for two main reasons:
>
> **1. More stable and selective learning:** In UGPP, the pre-trained GNN is fully frozen, meaning adaptation depends on aligning target samples with the source-trained manifold. Our proposed method, UGPrompt, does not treat all target samples equally; by using confidence-thresholded pseudo-labels, it focuses on samples that are already compatible with the source model. **As shown in our Maximum Mean Discrepancy (MMD) analysis** (see response to Reviewer euRC, W5), these confident samples are significantly closer to the original source distribution. This selectivity reduces harmful updates from heavily shifted samples. In contrast, supervised baselines train on all available labels (even those that may be strongly shifted), which can lead to overfitting or inconsistent adaptation directions.
>
> **2. Lower variance through full-distribution access:** Supervised baselines use a small labeled subset (e.g., 25%), which may lack the diversity needed to represent the target distribution under shift. In contrast, UGPrompt benefits from seeing the entire unlabeled target set, providing a more comprehensive view of the target manifold.
>
> **3. Supervised upper bound in few-shot settings:** To clarify that supervised signals do provide an upper bound when utilized effectively under shift, we refer to our few-shot experiments (Section 4.4, Table 4). While UGPrompt is primarily an unsupervised method, it can integrate labeled data when available. As Table 4 shows, giving UGPrompt access to labels consistently improves its performance over its fully unsupervised variant. This confirms that label supervision indeed acts as an upper bound, but only when the distribution shift is first effectively mitigated.
>
> Essentially, supervised signals can be harmful in this frozen setting if the labeled samples are poorly aligned with the source. UGPrompt avoids this by filtering for source-compatible samples and utilizing the full dataset. We will add this discussion to the revised version.
>
> >**W2: Justification for prompt tuning over other PEFT techniques like LoRA**
>
> **Response**
>
> We thank the reviewer for this valuable question. We chose input-level prompt tuning over structural PEFT methods like LoRA for two primary reasons:
>
> **1. Compatibility with Black-Box Models and Consistency Learning:**
> Our framework utilizes consistency regularization, comparing a weakly augmented graph with a strongly augmented/prompted version. By applying the prompt directly to the input, we can treat the pre-trained GNN as a black box. In contrast, LoRA injects low-rank matrices into specific internal layers. To implement our consistency loss with LoRA, we would need to manually route the weak augmentation through the original weights and the strong augmentation through the LoRA-modified layers. This would break the black-box assumption and requires architectural access to the GNN.
>
> **2. Parameter Efficiency for GNN Architectures:**
> While LoRA is highly effective for LLMs with massive weight matrices, standard GNNs typically have much smaller weight layers. Implementing LoRA across all layers would still introduce significantly more parameters than our current approach. Our method uses a very small number of trainable token vectors (fewer than 50 parameters across all experiments) applied only at the input. This minimal parameter count makes the model much less prone to overfitting, which is critical when adapting to unlabeled target datasets.
>
> Therefore, our approach is parameter-efficient, aligns with our input-driven augmentation pipeline, and inherits the black-box assumption. We will include this discussion in the final manuscript.
>
> >**C1: Notational inconsistencies**
>
> **Response**
>
> We thank the reviewer for pointing this out. We apologize and will fix the inconsistency of abbreviations.
>
> >**C2: Inconsistent citation formatting**
>
> **Response**
>
> We thank the reviewer for pointing this out. We apologize and will fix the citation inconsistency.

---

> > ### Comment · Reviewer_q58f · 2026-04-07
> >
> > Thank you for the detailed explanation, it really helped clarify several of the earlier questions. I do have a quick follow-up for further clarification.
> >
> > In your second response, you mention that the graph acts as a black-box model. If I understand this correctly, it would imply that the internal structure of the model is not accessible, and therefore neither the model parameters nor the gradients can be directly obtained.
> >
> > Given this, I'm a bit unclear on how prompt tuning is able to update its parameters. Specifically, how are gradients computed or approximated in this setting? Alternatively, does "black box" here refer more to a model-agnostic setup rather than a strictly inaccessible model?

---

> > > ### Author Response · Authors · 2026-04-07
> > > **Thank you!**
> > >
> > > We thank the reviewer for this precise follow-up and apologize for the confusion. We agree that an inaccessible model precludes gradient-based optimization. As the reviewer accurately framed in the alternative scenario, our framework is a **model-agnostic setup**.
> > >
> > >
> > > While we do require the gradients of the frozen base GNN to update the the external prompt parameters at the input level, our method is "agnostic" because it treats the pre-trained GNN as a static, unalterable functional block. We would like to clarify this further as follows. We used "black-box" in the initial response to highlight that UGPrompt requires no internal modifications, no knowledge of the GNN architecture, and no "rerouting" of data. However, we have not used the term in the paper. As explicitly phrased in **Section 2.2 (paragraph 3), Appendix A.6.3 (paragraph 1), and A.6.10 (paragraph 1)**, our problem definition (UGPP) and method (UGPrompt) are defined as a **model-agnostic setup**. This allows UGPrompt to be applied universally to any conventional pre-trained GNN simply by modifying the input feature matrix. In the final version, we will ensure that the model's role remain strictly aligned with the **model-agnostic** framing to avoid any ambiguity.

---

> > > > ### Comment · Reviewer_q58f · 2026-04-15
> > > >
> > > > I appreciate the authors' detailed response and have no further questions. I have provided my final recommendation.

---

### Review · Reviewer_AKeh · 2026-03-24

**Summary Of Contributions:**

The authors identify two weaknesses of existing domain adaptation and prompting methods on graphs, namely (i) their reliance on labeled data, and (ii) the need to train individual prediction heads for each downstream task. In light of this, they first propose the Unsupervised Graph Prompting Problem (UGPP) to define a distinct domain adaptation setting that alleviates the aforementioned issues. They then propose UGPrompt which operates in this setting to adapt a frozen GNN to data distribution shifts via consistency regularization and pseudo-labeling. Within the frozen-backbone restricted setting, UGPrompt outperforms other GNN prompting methods as well as setups that fine-tune the GNN projection heads under several distribution shift settings.

**Strengths:**
1. The authors first set a clear, well-defined problem formulation by defining UGPP and distinguishing it from existing formulations and methods, which is essential to support the rest of the work.
2. I think the motivation for the paper is sound and provides an interesting perspective on domain adaptation — instead of fine-tuning the GNN itself to adapt to distribution shifts, can we learn how to transform the new input data approach the source distribution while maintaining the task semantics?
3. The empirical results are strong, comprehensive and quite convincing: The authors show the superiority of their method under a wide array of tasks (node/graph classification) and distribution shifts (homophily, PageRank, class distribution shift and transfer learning settings on real-world datasets, density and clustering coefficient shifts in the appendix), outperforming a variety of both supervised baselines and SFDA methods. Additional ablation studies demonstrate that the method reliably generalizes across GNN backbones and even prompting functions, can leverage labels effectively when available, and provide useful insights into regularization and augmentation parameters and computational costs.

**Weaknesses:**
1. While I find the UGPP formulation and the analogy to LLM prompting useful, I think several clarifications are needed in how to position UGPP within existing lines of work.
   1. UGPP strikes me as a specific subset of SFDA assuming a more restrictive setting of having the source GNN fully frozen, thus requiring an “adapter head” in the form of a prompting function $f$ instead of fine-tuning the model or the prediction heads. With this in mind, I consider the current framing of UGPP as a distinct or even contrasting framework than SFDA somewhat misleading.
   2. I also generally disagree with the “GNN prompting” terminology. Prompting in LLMs/NLP relies on general few/zero-shot capabilities of the pre-trained model, which relatively small GNNs trained on orders of magnitude smaller datasets naturally lack. “Prompting” in our context is an adapter head trained to counteract distribution shifts here. However, I am aware that this terminology is not unique to this paper and is used in similar fashion in several works UGPrompt is compared against (such as the GPF-Plus paper the prompting function is based on), so I am not strongly opinionated on this point.
2. Some questions remain regarding the fairness of the results. Specifically, the GNN prompting baselines compared against are not allowed any fine-tuning, and thus are applied the same restrictions as UGPrompt. This is sensible from one perspective, but also implies that these methods are not evaluated on the setting they are designed for, and are at a comparative disadvantage. This doesn’t invalidate the results presented, but qualifies their strength somewhat.
3. UGPrompt has a strict inherent limitation that requires any distribution shift be over the same node/graph labels — this is clearly acknowledged by the authors and is an acceptable limitation, but still limits method applicability.

**Additional Comments:**

N/A

**Audience:**

Yes

**Audience Explanation:**

Yes, this will be particularly relevant to researchers in graph learning that are interested in OOD generalization, domain adaptation and transfer learning in addition to the more specific area of GNN prompting.

**Claims And Evidence:**

Yes

**Claims Explanation:**

Yes, I think this is arguably the key strength of the paper. While I may have my reservations regarding certain framing and evaluation decisions, the claims of the paper are very clearly stated and contrasted with prior work; just as importantly, as mentioned in S3, the authors provide ample evidence to support their arguments convincingly.

**Requested Changes:**

While I am leaning towards acceptance as-is, the proposed changes would strengthen the work. I deem (2) more important than (1) as it relates to the validity of the empirical evaluations, whereas (1) is somewhat more open to interpretation and is more concerned with the conceptual framing. I expect the typos (3), the list of which below are not comprehensive, to be fixed.
1. Re: W1.1: I suggest the authors review their positioning with respect to SFDA — I remain unconvinced that UGPP is a wholly distinct framework than SFDA, rather than a specific, more restricted variant of it.
2. Extending a subset of the evaluations to consider fine-tuned projection heads of appropriate prompting methods to represent their performance in the intended setting would allow for a fairer comparison. Even if these methods would match or outperform UGPrompt under more relaxed settings, the advantage of UGPrompt in the frozen-GNN setting remains — the goal here would be to make the evaluation more comprehensive, not to undermine the provided UGPrompt results.
3. There are several typos throughout the paper, an additional pass is required to rectify them:
   - Page 1, Introduction, first paragraph: “Subsequently, it…” The “it” refers to the aformentioned “first” and “second” challenges, in which case it should be replaced with “they”. In any case, this and the previous sentence does not flow well.
   - Page 1, Introduction, first paragraph, last line: “prompts by [~~the~~] human…”
   - Page 5, Sec. 3.2: $\mathcal{L}_d$ is used in the text to refer to the loss optimized in (4), but this is not used in (4) itself — though upon reading this becomes clear but I suggest (4) is adjusted accordingly.
   - Page 6, Sec 3.3: “[~~the~~] Appendix A.6.6”
   - Page 6, Sec 3.3, last line: “UGPrompt is more efficient [than -> on] average at the [~~test~~] time”
   - Page 9, Sec. 4.3.2: “Higher $p_s$ yields better performance[~~,~~]”: The comma should be omitted.
   - Page 10, Sec. 4.5: The reference to Table 6 in text is not linked to the table.
   - Tables 4 and 5 should switch — Table 5 is in age 10 but is first referred to in page 9 (Sec. 4.3.3), Table 4 is in page 9 but is first referred to in Sec. 4.4.

---

> ### Author Response · Authors · 2026-04-03
> **Response to the Reviewer - Part I**
>
> >**W1.1: Framing of UGPP relative to source-free domain adaptation.**
>
> **Response**
>
> We thank the reviewer for this constructive feedback. We agree that UGPP can be viewed as a restrictive, prompt-based subset of the broader Unsupervised Source-Free Domain Adaptation (USFDA) framework.
>
> In the paper, we intend to highlight two distinct advantages of this restrictive setting over conventional graph USFDA: (i) We isolate performance gains of the prompting function from latent fine-tuning of the encoder or prediction heads. (ii) UGPP treats the GNN as a black box, making it uniquely compatible with inaccessible models or privacy constraints. However, we agree that our current wording may overstate this distinction.
>
> To address this, we will soften the language in the revision to better position UGPP within the USFDA landscape. For example, we will remove terms like "fundamentally differs" (Section 2.2) and explicitly state that UGPP acts as a prompt-based (restricted) setting of the USFDA.
>
> Furthermore, to ensure a comprehensive evaluation in this context, we have compared UGPrompt with two recent graph-based SFDA methods (SOGA and GraphCTA) in **Appendix A.6.4 (Table 14)**. As shown, UGPrompt outperforms these fine-tuning approaches, validating the effectiveness of our prompting.
>
> >**W1.2: Terminology of graph prompting versus traditional feature-space adapters.**
>
> **Response**
>
> We appreciate the reviewer’s perspective. We agree that current GNNs lack the scale and inherent zero-shot reasoning capabilities of LLMs. However, the use of the term "prompting" in our work (and in the baselines) refers to the *adaptation paradigm* rather than the scale of the model.
>
> Specifically, our framework directly mirrors the standard "Pre-train, Prompt, and Predict" pipeline established in NLP [1]. By enforcing task unification, pre-training and weight freezing, and training only external, input-level tokens, we adopt the LLM prompting mechanism.
>
> We view parameter-efficient GNN prompting as a necessary step toward future Graph Foundation Models (GFMs). As we discussed in the Introduction, building GFM at the level of LLMs has two big challenges: the lack of universal task formats (like next-token prediction) and the limited reasoning capability of GNNs. Therefore, our work, particularly because of our restrictive problem definition and, more importantly, its unsupervised nature, establishes the methodological groundwork for adapting future large-scale GFMs in a privacy-preserving and parameter-efficient manner.
>
> [1] "Pre-train, Prompt, and Predict: A Systematic Survey of Prompting Methods in Natural Language Processing," ACM Comput. Surv., 2023.

---

> ### Author Response · Authors · 2026-04-03
> **Response to the Reviewer - Part II**
>
> >**W2: Evaluation fairness regarding baseline restrictions.**
>
> **Response**
>
> We appreciate the reviewer’s perspective on the fairness of the comparison. While baselines were implemented using their original source code and settings, we restricted their decoders to align with the UGPP requirement that the entire pre-trained model remains frozen. This constraint is necessary to isolate the prompt's specific impact and avoid conflating it with head fine-tuning.
>
> Now, to show how UGPrompt compares to supervised methods in their standard unconstrained training regime, we conducted a new experiment where we allowed the supervised baselines to **fine-tune their decoders** using 25% of the labeled target data. As shown in **Table R1**, even with the advantage of fine-tuning their decoders on labeled data, the baselines frequently suffer from severe negative transfer under distribution shift. In contrast, our fully unsupervised, frozen-head UGPrompt achieves highly stable, positive improvements and outperforms the baselines in the majority of cases.
>
> (GPPT is not applicable to graph classification, so we report NA for that)
>
> **Table R1: Comparison of Improvement of performance (IMP%) against baselines with fine-tuned decoders (25% labels).**
> |Method|CIT|CNL|COR|DHFR|ENZ|PRO|PUB|TEX|WIS|
> |-|-|-|-|-|-|-|-|-|-|
> |All-in-One|-19.8|-34.6|-10.6|-0.4|-24.2|-25.5|-10.0|-11.9|-36.5|
> |GPF-Plus|-2.9|5.8|-1.6|2.1|-17.4|-17.8|10.3|25.0|4.8|
> |GPPT|-12.5|-20.9|-13.6|NA|NA|NA|-9.8|9.3|-7.9|
> |GraphPrompt|-3.6|-5.2|-0.7|-6.0|-21.3|-1.9|-0.5|23.3|-11.9|
> |GraphPromptPlus|-9.6|12.0|-9.0|-14.7|-49.3|-13.7|8.7|27.5|-0.8|
> |UGPrompt (ours)|1.3|23.6|2.5|1.4|0.8|6.9|6.1|13.6|12.7|
>
> >**W3: Applicability limitations due to the requirement for shared label spaces.**
>
> **Response**
>
> We agree that the requirement for consistent label sets is a limitation due to the frozen projection head necessary for the UGPP setting. This reflects a broader challenge in the field that, unlike NLP, which operates on a universal vocabulary (token space), graphs currently lack a unified discrete output space. This makes generalizing to new classes difficult without retraining or fine-tuning a task-specific projection head.
>
> Despite this, we conducted an exploratory test (**Section 4.5**) on a restricted case of category shift in which the source and target domains had disjoint semantic concepts but the same number of classes. In this setup, labels were manually remapped while the decoder remained frozen. We found that UGPrompt still outperformed the baselines, even when the original semantic mapping of the frozen head was broken.
>
> Developing universal graph vocabularies to enable true, open-vocabulary prompting is a key direction for our future research.
>
> >**C3: Typos**
>
> **Reponse**
>
> Thank you for detecting the typos; we will fix them in the final version and incorporate your other suggestions.

---

> ### Comment · Reviewer_AKeh · 2026-04-30
>
> I thank the authors for the detailed response; I particularly appreciate the additional study in response to W2. I think the authors have also done a good job in distinguishing their work from SFDA, but also acknowledging the limitations of the work. I have submitted my final decision accordingly.

---

### Decision · Action_Editor_puem · 2026-05-17

**Recommendation:** Accept as is

**Audience:**

Yes

**Audience Explanation:**

The topic is relevant to an audience working on GNNs and the reviewers judged it pertinent unanamously.

**Claims And Evidence:**

Yes

**Claims Explanation:**

The paper proposes a new problem: unsupervised, source free prompt only based GNN adaptation. The authors propose a method based on pseudo-labeling and consistency regularizer to address this. Evaluations are provided on a number of relevant graph tasks. The reviewers were generally positive and found a few conceptual experimental and framing issues regarding the claims that were clarified in the revision. Thus the main claims of the paper are well supported.